# Ultrasound neuromodulation reveals distinct roles of the dorsal anterior cingulate cortex and anterior insula in learning

Nomiki Koutsoumpari[1,2]☯*, Johannes Algermissen[3,4]☯, Siti Nurbaya Yaakub[1,2], Hanneke EM den Ouden[4], Nadege Bault[1,2]☯, Elsa Fouragnan🄳[1, 2]☯*

1 School of Psychology, University of Plymouth, Plymouth, United Kingdom, 2 Brain Research Imaging Center (BRIC), University of Plymouth, Plymouth, United Kingdom, 3 Department of Experimental Psychology, University of Oxford, Oxford, United Kingdom, 4 Radboud University, Donders Institute for Brain, Cognition and Behaviour, Nijmegen, The Netherlands

☯ These authors contributed equally to this work.
* nomiki.koutsoumpari@plymouth.ac.uk (NK); elsa.fouragnan@plymouth.ac.uk (EF)

## Abstract

Pavlovian biases reflect how evolutionarily hard-wired tendencies—automatic approach toward reward cues and withdrawal from threat cues—can interfere with flexible, goal-directed action. Such biases arise through three mechanisms: (a) anticipated rewards energize action while anticipated punishments suppress it (response bias), (b) agents learn differently from actions than from inactions (learning bias), and (c) reward/punishment cues themselves drive repetitive behavior, independent of outcomes (perseveration bias). The neural origin of these biases is unclear. Past evidence suggests dorsal anterior cingulate cortex (dACC) and anterior insula (aIns) as part of a "reset network" that rapidly responds to salient information and might contribute to these biases. We used transcranial ultrasonic stimulation (TUS) in 29 healthy participants to interfere with neural activity in these regions and test their causal role in a within-subject, counter-balanced design across three sessions (sham, TUS-dACC, TUS-aIns). Computational modeling revealed a functional differentiation of both regions in Pavlovian biases: while TUS to either region did not affect the response bias, TUS to the aIns decreased people's learning bias, while TUS to dACC increased participants' perseveration bias. Although the dACC and aIns are part of the same network and often co-activate during decision-making tasks, TUS interference reveals their distinct roles: the dACC mediates cue-dependent persistence while the aIns is critical for inferring whether outcomes are self-caused.

## Introduction

Some of our everyday decisions are remarkably fast: in the blink of an eye, we paralyze before our hand reaches a potentially venomous spider, or we instinctively reach for the last remaining cookie before our siblings can grab it. Psychology and

License, which permits unrestricted use, distribution, and reproduction in any medium, provided the original author and source are credited.

**Data availability statement:** Data availability statement: The data files are available as .csv and .mat files on the OSF repository https://doi.org/10.17605/OSF.IO/PUX6S. Code availability statement: Analyses code to reproduce the results from regression and reinforcement learning models are in a GitHub repository under https://github.com/johalgermissen/mgng_tus_dacc_ains as well as on Zenodo under https://doi.org/10.5281/zenodo.19386479.

**Funding:** NK received internal pump-priming funds from the School of Psychology and the Brain Research Imaging Centre, University of Plymouth, in support of this study. EF received funding from a UKRI Future Leaders Fellowship grant: https://www.ukri.org/opportunity/future-leaders-fellowship-round-11/, Biotechnology and Biological Sciences Research Council: https://www.ukri.org/opportunity/bbsrc-standard-research-grant-2026-round-one-applicant-led-mode/, Neuromod+: https://neuromodplus.org/, ARIA: https://aria.org.uk/ (MR/Y034368/1 to EF), a Biotechnology and Biological Sciences Research Council grant (BB/Y001494/1 to EF), a Neuromod+ grant (EP/W035057/1 to EF), and an ARIA grant (SCNI-PR01-P15 to EF). The funders had no role in study design, data collection and analysis, decision to publish, or preparation of the manuscript.

**Competing interests:** The authors have declared that no competing interests exist.

**Abbreviations:** aIns, anterior insula; CEM43, Cumulative Equivalent Minutes at 43 °C; dACC, dorsal anterior cingulate cortex; EEG, electroencephalography; fMRI, functional magnetic resonance imaging; ISPPA, spatial peak pulse average intensity; ITRUSST, International Transcranial Ultrasonic Stimulation Safety and Standards consortium; MItc, transcranial Mechanical Index; MRI, magnetic resonance imaging; MRS, magnetic resonance spectroscopy; pseudo-CT, pseudo-computed tomography; TMS, transcranial magnetic stimulation; TUS, transcranial ultrasonic stimulation.

neuroscience have explained our ability for such fast, instinctive decisions by postulating the existence of multiple decision-making systems that trade-off speed against deliberation and sophistication [1–3]. One class of especially fast, but also rigid and seemingly hardwired decisions has been proposed as being driven by a Pavlovian control system [4–7]. This system rigidly responds with invigoration to any chance to gain rewards ("Go"), but with inhibition to any looming threat of punishment ("NoGo"). This bias has classically been called the Pavlovian *response bias* [8,9].

The Pavlovian control system gives rise to two other subtle, yet well-documented Pavlovian biases that influence learning. One the one hand, a Pavlovian *learning bias* describes people's tendency to overly claim rewards as caused by their actions, but ignore punishments that follow inaction. For example, a football coach might credit himself for his team's win after having changed the player lineup, but fail to acknowledge how not changing it led to repeated defeats in the past. One the other hand, a Pavlovian cue-valence dependent *persistence bias* describes people's tendency to interpret cues that signal the chance for rewards/punishments as reinforcer signals in themselves. In case of no (or an ambiguous) feedback, people are likely to persist in situations in which a reward could be obtained, but likely to change their behavior in situations in which a negative outcome might occur [10]. Pavlovian biases such as these become evident in tasks that orthogonalize action requirement (Go/NoGo) against the valence of potential outcomes (rewards/punishments) [8,9] and appear to be evolutionarily ancient and shared across the animal realm [11]. By shaping how we assign value to actions and learn from outcomes, these biases play a fundamental role in adaptive behavior, and their dysregulation may underlie core symptoms across a range of psychiatric disorders [12,13].

Pavlovian biases have been linked to neural activity across various cortical and subcortical regions [8,10,14], yet the precise causal contribution of each of these regions remain unclear. Pinpointing when and where in the brain these biases emerge is essential for uncovering their underlying mechanisms. We have recently observed that the mere anticipation of potential punishment elicits a rapid freezing of gaze within 200–300 ms of cue onset [15], coinciding with early value-related components in electroencephalography (EEG) recordings [10,14]. These findings suggest that value-related information is processed remarkably early, allowing it to exert a fast and automatic influence on action selection. EEG-informed functional magnetic resonance imaging (fMRI) analyses have further revealed that such early neural responses recruit a network involving the dorsal anterior cingulate cortex (dACC) and anterior insula (aIns), which are selectively engaged by salient/negative information and appear to support rapid behavioral adaptation [16]. In contrast, later feedback processing stages engage the striatum to support more deliberate reinforcement learning. Together, these findings position the dACC and aIns as strong candidates for causal contributors to Pavlovian biases observed in behavior.

Selective interference with neural processing in these deep cortical regions to test their role in Pavlovian biases is exceedingly challenging with conventional noninvasive brain stimulation techniques which are limited to superficial cortical regions. However, recently, transcranial ultrasonic stimulation (TUS) has emerged as a

promising non-invasive tool for neuromodulation [17,18] that can stimulate deep brain regions with unprecedented spatial precision when careful precautions are taken to limit the transmission loss caused by the skull. TUS has been successfully used to change choice behavior and neural activity in non-human primates [19,20] and humans [21,22]. We thus tested the role of dACC and aIns in Pavlovian biases using a single-blinded within-subject counterbalanced 3-session (dACC/aIns/sham) offline TUS experiment.

Given that both regions form part of an early "reset" network responding to salient stimuli [16] and show rapid valence-related responses [10,14], we predicted that TUS to either region would affect one or more Pavlovian biases. We predicted that TUS to either region would selectively affect action-specific Pavlovian biases, in particular the Pavlovian response bias, defined as the tendency to invigorate action in response to rewards and suppress action in response to punishments. Additionally, based on dACC's role in behavioral adjustment and processing counterfactual feedback [20,23] and on the aIns's role in credit assignment and distinguishing self-caused from externally-caused outcomes [24,25], we predicted that perturbing either regions could plausibly alter Pavlovian influences on action selection and learning. We therefore examined whether TUS to dACC or aIns would differentially affect distinct components of Pavlovian control. To this end, we used computational reinforcement learning models designed to isolate these three distinct Pavlovian biases and test our predictions.

We expected TUS to interfere with neural processing in these regions and thus lead to different expression of Pavlovian biases in behavior [8,9] since both regions have been implicated in different aspects of feedback-based learning, including biased credit assignment [9,10,26]. We thus also tested for effects on several Pavlovian biases, including a *persistence bias* and a *learning bias,* using computational reinforcement learning models designed to isolate latent cognitive processes.

To foreshadow our results, stimulation of both dACC and aIns altered Pavlovian biases, revealing functional differentiation between these regions. TUS-aIns selectively reduced participants' tendency to overly attribute rewards to their own actions and their reluctance to attribute punishments to their inactions, captured by a learning bias parameter [9]. In contrast, TUS-dACC impaired participants' ability to tell apart the valence of the feedback from the valence of the cue they responded to, captured by a persistence bias parameter [10]. These results indicate dACC relative dominance in selecting which cues in the environment to use as a reinforcer signal, and of the aIns in inferring whether outcomes are self-caused. These two mechanisms—reliance on cue valence as a reinforcer signal and assigning credit to one's own actions—may be altered in psychiatric disorders [27,28] potentially contributing to symptoms such as pathological guilt, externalized blame, and compulsive repetition of actions despite unhelpful outcomes. Disentangling their specific neural contributions is therefore essential for advancing both cognitive neuroscience and clinical psychiatry.

This study also demonstrates the transformative potential of TUS for causally probing decision-making processes in deep cortical regions typically beyond the reach of conventional non-invasive techniques such as transcranial magnetic stimulation (TMS). By targeting the dACC and aIns, two regions often grouped within the salience network and previously implicated in rapid valence-action processing, we show that they play dissociable roles in biased learning. The ability to modulate them independently with TUS represents not only a significant methodological advance but also lays the foundation for mechanistically precise interventions targeting maladaptive decision-making in psychiatric disorders.

## Results

### Participants and procedure

Twenty-nine participants, screened for any counter indications to TUS and magnetic resonance imaging (MRI) (S2 Appendix) completed a four-session study protocol. The first session involved collecting MR images for neuronavigation and generating a pseudo-computed tomography (pseudo-CT) image for personalized acoustic planning [29] as well as a task practice (Fig 1A). Participants were then assigned to one of three TUS conditions: Sham, dACC, or aIns (Fig 1B), using a counterbalanced design. Each of these subsequent sessions included an 80-s-long 5 Hz-patterned TUS intervention,

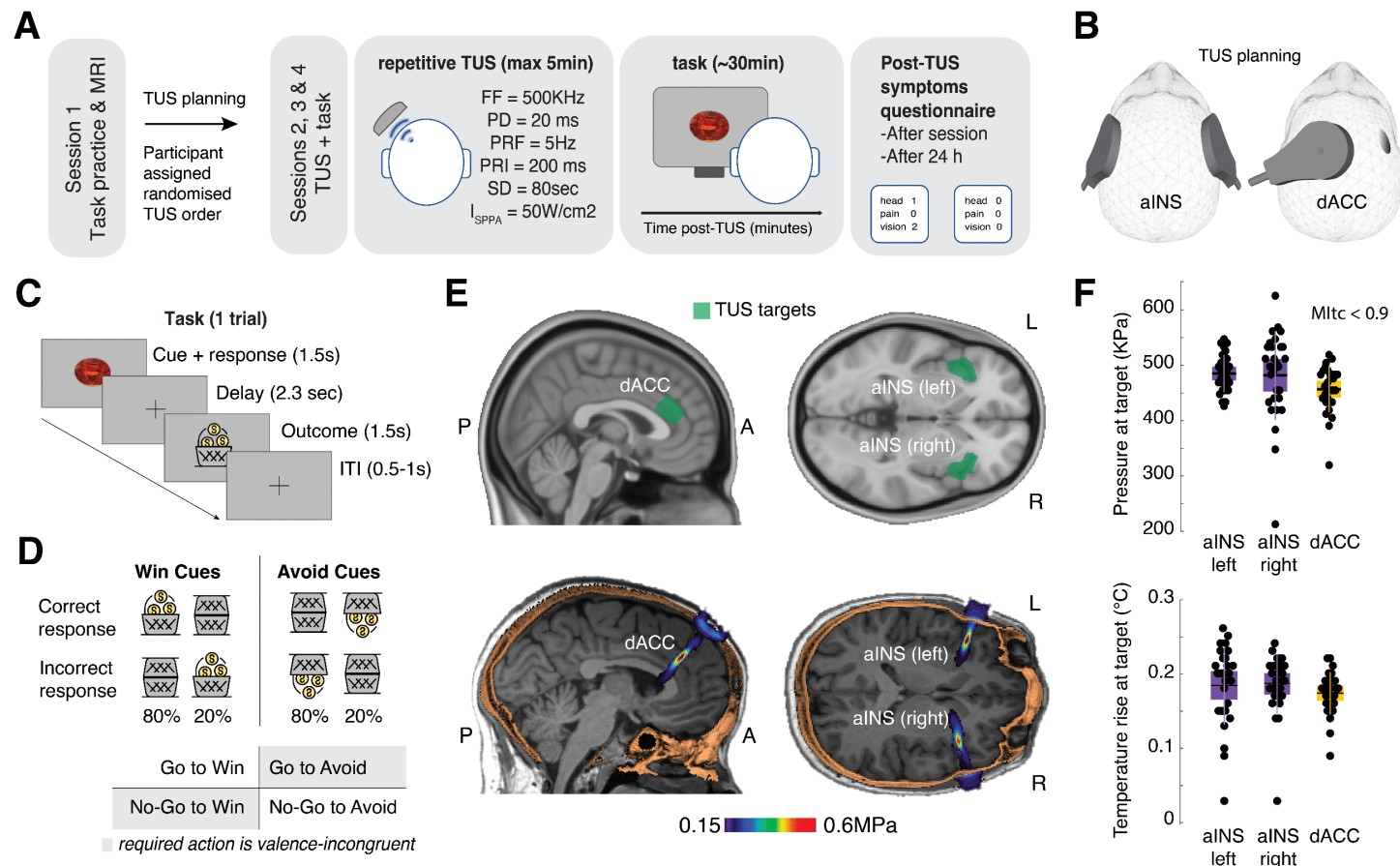

**Fig 1. Study design, TUS targeting, task structure, and acoustic simulations. A. Study Design:** In the first session, a T1-weighted image was acquired for neuronavigation and acoustic planning. Participants also practised a short version of the task. In sessions two to four, participants were randomly allocated to one of three conditions: Sham, dACC, or bilateral aIns. Post-stimulation safety questionnaires were completed on the same day and the following day. **B. TUS positioning:** Visualization of the positioning of the transducer on the head for the active TUS sessions. **C. Task design:** On each trial, participants saw an abstract cue and needed to learn from trial-and-error to respond with either a Go or NoGo response to that cue. The cue disappeared at the response onset, followed by a jittered interval and finally response-dependent feedback. Each cue had one correct action (Go or NoGo) and was either a Win cue (leading to rewards or neutral outcomes) or an Avoid cue (leading to neutral outcomes or punishments). Rewards and punishments were represented as money falling into or out of a bucket. **D. Feedback validity and cue types:** Each block of session included 4 novel cues, varying in valence (Win/Avoid) and required action (Go/NoGo). Incongruent cues, where the required action was in opposition to the Pavlovian bias (NoGo to Win, Go to Avoid punishment), are highlighted in gray. Feedback was probabilistic. Correct responses to Win cues resulted in rewards 80% of the time and neutral outcomes 20% of the time. Correct actions to Avoid cues led to neutral outcomes in 80% of cases and punishments in 20%. Incorrect responses led to feedback with the reverse probabilities. **E. Regions:** Targeted brain regions for TUS (dACC and bilateral aIns) are shown (top), with post-stimulation pressure simulated through k-Plan software (BrainBox) expressed in megapascals on a representative participant (bottom). **E. Regions:** Targeted brain regions for TUS (dACC and bilateral aIns) are shown (top), with post-stimulation pressure simulated through k-Plan software (BrainBox) expressed in megapascals on a representative participant (bottom). **F. TUS simulations:** Maximum pressure in the brain target volume (top panel) and maximum temperature rise in the brain target volume (bottom panel), (*n* = 29). Box plots show the mean, and the standard error (bounds of the box). Data from each individual participant are presented as small black circles. The data underlying this figure can be found in S1 Data.

with a spatial peak pulse average intensity (ISPPA) of approximately 50 W/cm² in water, immediately followed (maximal latency: 10 min) by a motivational Go/NoGo learning task [8–10]. While the task structure remained consistent across sessions, the cue sets were different and individually randomized to prevent learning effects. Each session lasted approximately 1.5 hours.

During the motivational Go/NoGo learning task, on each trial, participants were presented with one of several cues. They had to learn from probabilistic feedback via trial-and-error to either perform a Go (button press) or a NoGo action (no button press) to four cues per block. Half of these cues were Win cues for which correct responses (mostly) returned a reward (increase in point score), while incorrect responses (mostly) led to neutral feedback (no change in point score). For the other half of cues, called Avoid cues, correct responses (mostly) led to neutral feedback, while incorrect responses (mostly) led to punishments (decrease in point score). The task timeline is presented in Fig 1D. Feedback was probabilistic such that correct responses resulted in the desired outcome (reward or omitted punishment) 80% of the time, whereas incorrect actions led to the desired outcome in only 20% of trials. On each session, participants completed 320 trials, split into four blocks of 80 trials. In each block, a distinct set of four cues was presented, each repeated 20 times. The order of cue presentations was randomized for each participant within each block (Fig 1D).

Targeting of the bilateral aIns and dACC was individually adjusted using T1-weighted MRI (structural scans) during planning for target nd transducer placement, as well as acoustic and thermal simulation. The dACC was defined anatomically as the region anterior to the anterior commissure, dorsal to the genu of the corpus callosum, and bounded dorsally by the cingulate sulcus. The aIns was identified as the anterior portion of the insular cortex, located deep within the Sylvian fissure, anterior to the central insular sulcus, and bordering the frontal operculum. S1 Fig shows a representative participant for the region targeted. Repetitive TUS followed a 5 Hz-patterned protocol with a 10% duty cycle (Fundamental Frequency = 500 kHz, Pulse Duration = 20 ms, pulsed every 200 ms), applied for 80 s (400 pulses in total). The ISPPA in water was maintained at 50 W/cm$^2$ across participants (validated in a hydrophone tank [22]). We took care to remain within guidelines for human ultrasound exposure as defined the International Thermal and Radiological Ultrasound Safety Standards and Thresholds (ITRUSST; [30]). Importantly, the transcranial Mechanical Index (MItc) was way below 1.9 and the results of our acoustic simulations predominantly indicated a maximum soft tissue temperature rise below 2 °C (S1 Table). We also calculated the Cumulative Equivalent Minutes at 43 °C (CEM43), a metric reflecting both duration and intensity of heating relative to 43 °C, the critical threshold for thermal cell damage. The CEM43 values were always below 0.1, i.e., well below the safety threshold of 0.25 [30].

Sham was administered in the same manner as active aIns-TUS, placing the transducer over the site but without actual stimulation. Instead, participants listened to a sound that mimicked the TUS protocol's acoustic output (validated across 6 participants independent from this study), played through bone conduction headphones. Acoustic simulations for each participant were used to plan the stimulation to ensure accurate pressure delivery at the target area and adherence to safety protocols (an example participant's targeted areas are shown in Fig 1E). Detailed acoustic simulation parameters and outputs for all study participants can be found in Fig 1F, S1 and S2 Tables, and are summarized in Table 1.

**Table 1. Ultrasound parameters across different target tissues for dACC and aIns stimulation. A sphere (Target ROI) with a 10mm radius centered on the target coordinates is used to reliably extract peak values from the brain region where the focus is located.**

| Target | Tissue | Temperature (°C) | | Pressure (MPa) | | ISPPA (W/cm$^2$) | | MItc | |
|--------|--------|------|------|------|------|------|------|------|------|
| | | *M* | *SD* | *M* | *SD* | *M* | *SD* | *M* | *SD* |
| **dACC** | Target ROI | 37.18 | 0.05 | 0.45 | 0.04 | 6.69 | 1.16 | 0.63 | 0.06 |
| | Soft tissue | 37.96 | 0.28 | 0.45 | 0.04 | 6.72 | 1.18 | 0.63 | 0.06 |
| | Skull | 38.73 | 0.41 | 0.32 | 0.08 | 3.57 | 2.29 | 0.45 | 0.12 |
| **l-aIns** | Target ROI | 37.19 | 0.04 | 0.47 | 0.08 | 7.60 | 2.26 | 0.67 | 0.11 |
| | Soft tissue | 38.14 | 0.34 | 0.48 | 0.06 | 7.74 | 1.98 | 0.68 | 0.09 |
| | Skull | 38.99 | 0.64 | 0.33 | 0.05 | 3.63 | 1.09 | 0.46 | 0.07 |
| **r-aIns** | Target ROI | 37.17 | 0.03 | 0.48 | 0.03 | 7.53 | 1.06 | 0.67 | 0.05 |
| | Soft tissue | 38.20 | 0.38 | 0.48 | 0.03 | 7.54 | 1.05 | 0.67 | 0.05 |
| | Skull | 39.10 | 0.62 | 0.35 | 0.06 | 4.06 | 1.39 | 0.49 | 0.08 |

## Side effects

Participants completed a post-TUS questionnaire for side effects twice: once at the end of each session, and a second time 12–48 hours after the session. They were presented with a list of symptoms for which they had to rate the intensity of the symptom and the degree to which they believe that symptom was related to the stimulation (refer to the Methods section under TUS protocol and procedure for further safety details; see also S1 Appendix for the full safety questionnaire). Immediately after stimulation of the insula, participants reported the following side effects: anxiety ($N=1$), difficulty paying attention ($N=2$), forgetfulness ($N=1$), headache ($N=1$), and sleepiness ($N=3$). All side effects were rated as mild and unrelated to the stimulation. Following stimulation of the dACC, moderate anxiety ($N=1$), mild forgetfulness ($N=1$), moderate neck pain ($N=1$), mild to moderate sleepiness ($N=2$), and mild vision problem ($N=1$) were reported. Only the participant reporting vision problems considered those as possibly related to the stimulation. Following sham stimulation, one participant reported mild itchiness ($N=1$) and another moderate sleepiness ($N=1$), again both problems were rated as unrelated to the stimulation. Overall, no qualitative difference in the number and quality of side effects was observed between sham and active stimulation. In most cases of side effects, participants did not attribute them to the stimulation.

## Pavlovian biases in responding and learning in the sham session

We first focus on the data from the sham session to replicate Pavlovian biases in responding and learning as typically observed in this task [8–10].

First, in line with previous work [9,10,31,32], we used mixed-effects logistic regression to fit Go/NoGo responses as a function of the required action of a given cue (Go versus NoGo), the cue valence (Win versus Avoid cues), and their interaction. Participants successfully learned the task, as they made more Go responses to Go than NoGo cues (main effect of required action: $b=2.132$, 95%-CI [1.797, 2.467], $\chi^2(1) = 53.812$, $p<.001$; Fig 2D). Furthermore, there was a Pavlovian bias in behavior, indexed by significantly more Go responses to Win than Avoid cues (main effect of cue valence: $b=0.689$, 95%-CI [0.421, 0.966], $\chi^2(1) = 17.173$, $p<.001$). Lastly, the interaction between both terms was significant ($b=0.269$, 95%-CI [0.104, 0.434], $\chi^2(1) = 9.646$, $p=.002$), reflecting a stronger Pavlovian bias for Go than for NoGo cues. Similar analyses of the reaction times (RTs) also revealed a bias (S2 Fig). These analyses confirm that participants' behavior was influenced by Pavlovian biases in the sham session, which constitutes a prerequisite for modulating them in the other two active sonication sessions.

Second, we used mixed-effects logistic regression to analyze participants' propensity to repeat versus switch a response on the next trial as a function of the obtained outcome and performed response on the current trial. We have previously found that Pavlovian biases are present not only in action selection, but also in learning, with a larger propensity to take valenced feedback into account (i.e., repeat actions after rewards, switch actions after punishments) for Go actions compared to NoGo actions. Specifically, learning is increased for rewarded Go (relative to rewarded NoGo) actions and decreased for punished NoGo (relative to punished Go) actions (see detailed explanation in the computational modeling) [9,10]. Because this bias should only occur for valenced outcomes (rewards, punishments), but not for neutral outcomes, we only included valenced outcomes in this analysis. We observed a significant main effect of the obtained outcome (reward versus punishment), $b=1.436$, 95%-CI [1.190, 1.683], $\chi^2(1) = 50.170$, $p<.001$, with more response repetitions after rewards than after punishments, and a main effect of performed response, $b=0.211$, 95%-CI [0.081, 0.341], $\chi^2(1) = 6.422$, $p=.001$, with more response repetitions for Go than NoGo responses (likely due to the fact that participants showed overall more Go than NoGo responses in this task). Crucially, the interaction between the obtained outcome and performed response was significant, $b=0.423$, 95%-CI [0.195, 0.650], $\chi^2(1) = 10.051$, $p<.001$, with a stronger outcome effect after Go than after NoGo responses (Fig 2E). This result confirmed the presence of Pavlovian biases also in learning from feedback.

Thirdly, we used mixed-effects logistic regression to analyze participants' propensity to repeat versus switch a response on the next trial as a function of the cue valence. We have previously found that participants show a propensity to repeat responses for Win cues than for Avoid cues, suggesting that they also partly interpret the cue valence as a feedback

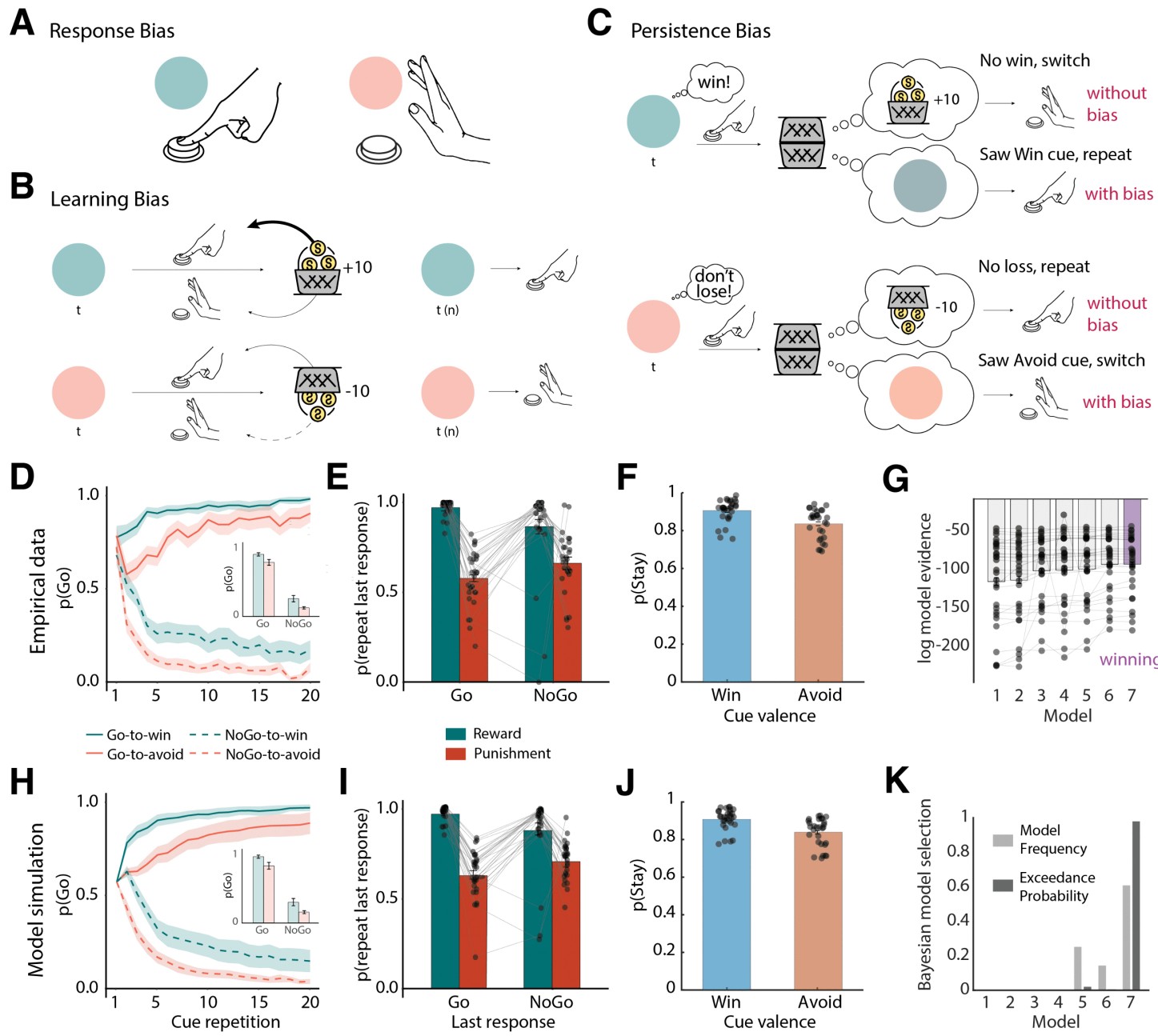

**Fig 2. Behavioral performance in the sham dataset. A. Response bias:** Humans exhibit a tendency to show actions (i.e., button presses) for Win cues (blue), aligning reward seeking with action invigoration, and to withhold actions for Avoid cues (red), aligning punishment avoidance with action suppression/inaction. **B. Learning bias:** Learning is enhanced for rewarded Go actions compared to rewarded NoGo actions, promoting faster acquisition of Go actions from rewards. Conversely, learning is reduced for punished NoGo actions compared to punished Go actions, slowing the extinction of NoGo responses via punishments. **C. Persistence bias:** Seeing a Win cue partly acts as a reward in itself, leading to a higher tendency to repeat responses to such cues, even when a neutral outcome (negative feedback) suggests that one should switch the response. In contrast, seeing an Avoid cue partly acts as a punishment in itself, leading to a higher tendency to switch responses, even if a neutral outcome (positive feedback) suggests repeating the response. **D.** Trial-by-trial proportion of Go responses (error bands = ± SEM, n = 29) for Go cues (solid lines) and NoGo cues (dashed lines). A clear response bias is evident from the start, with participants making more Go responses to Win (green lines) than Avoid (red lines) cues. Participants show a general bias towards Go responses, with initial response rates around 80%. Additionally, instrumental learning is apparent in Go responses increasing for Go cues and decreasing for NoGo cues over time. **E.** Probability to repeat a response on the next encounter of the same cue as a function of the response and outcome for valenced outcomes (rewards, punishments) only (error bars are ± SEM across participants, n = 29, dots indicate

individual participants). Learning was reflected in the higher probability of staying after rewarded responses than after being punished (main effect of outcome valence). Biased learning was evident in the stronger outcome effect after Go responses than after NoGo responses, indicating the presence of a learning bias. **F.** Probability to repeat a response on the next encounter of the same cue as a function of cue valence (error bars are ± SEM across participants, $n = 29$, dots indicate individual participants). Participants showed a higher tendency to repeat responses for Win than Avoid cues, indicative of a *persistence* bias. **G.** Laplace approximation to the log-model evidence favors the model incorporating all three Pavlovian biases—response bias, learning bias, and persistence bias (M7)—over simpler models (M1–M6); error bars are ± SEM across participants, $n = 29$. **H–J.** Posterior predictive checks: one-step-ahead predictions based on the best fitting parameters (estimated with hierarchical Bayesian inference) for M7. **K**. Bayesian model selection. The winning model is M7 as determined by model frequency and protected exceedance probability, in line with log model evidence. The data underlying this figure can be found in S2 Data.

signal [10]. This leads to response repetition (persistence) tendencies that depend on the cue valence, a tendency that was not well captured by previous computational models (S4 Fig), but can only be accounted for by a third, *persistence bias*. Indeed, we found a strong main effect of cue valence on response repetitions, $b = 0.351$, 95%-CI [0.262, 0.440], $\chi^2(1) = 60.117$, $p < .001$, with more repetitions for Win than Avoid cues (Fig 2F). This result indicated the presence of a third Pavlovian bias in which the cue valence affected persistence tendencies. Taken together, these results motivated us to further interrogate these biases using computational reinforcement learning models.

As control analyses, we tested for age or gender effects on any of the reported effects. For Go/NoGo responses, there were significant interaction between age and required action ($b = -0.415$, 95%-CI [−0.685, −0.145], $\chi^2(1) = 9.061$, $p = .003$) and gender and required action ($b = -0.293$, 95%-CI [−0.573, −0.013], $\chi^2(1) = 4.206$, $p = .040$), reflecting effects on task accuracy: Male participants achieved overall higher accuracy than female participants, and younger participants higher accuracy than older participants. For repeat/switch responses, there was a main effect of gender ($b = -0.276$, 95%-CI [−0.515, −0.038], $\chi^2(1) = 5.157$, $p = .023$) as well as significant interaction between gender and previous outcome ($b = -0.310$, 95%-CI [−0.528, −0.092], $\chi^2(1) = 7.743$, $p = .005$): Male participants showed overall more repetitions than female participants as well as a stronger effect of past feedback. Specifically, male participants more often repeated responses after positive feedback, which might have contributed to higher performance. Furthermore, there was a significant main effect of age ($b = -0.313$, 95%-CI [−0.540, −0.085], $\chi^2(1) = 7.250$, $p = .007$), with older participants repeating responses less often than younger participants. This propensity to more often switch responses might have contributed to lower accuracy in older participants.

As a further control analyses, we tested for effects of session identity (1st, 2nd, 3rd session, using data from all sonication conditions to increase power) on the reported effects. For Go responses, there was an interaction between session ID and required action ($\chi^2(1) = 18.189$, $p < .001$), reflecting that accuracy increased over sessions. For repeat/switch responses, there was a significant main effect of session ($\chi^2(1) = 18.916$, $p < .001$) and a significant interaction between session and past outcome ($\chi^2(1) = 7.898$, $p = .019$): participants overall repeated their responses more often in later sessions and showed a stronger effect of past outcomes on choice repetitions, mostly driven by more repetitions after positive feedback. This pattern might have contributed to higher accuracy in later sessions. We furthermore tested for effects of session order (i.e., order in which all three stimulation conditions were performed) and effects of cue set (i.e., the set of geometric cues which participants saw in a given block), but did not find any effects on Go/NoGo or repeat/switch responses.

## Computational modeling of the sham data

We next fit a series of increasingly complex reinforcement learning models to participants' responses in the sham sessions. These models follow previous publications on this task [8–10]. These models incorporate three previously described forms of Pavlovian biases: (a) a response bias, (b) a learning bias, and (c) a persistence bias.

The first bias, called ***response bias***, is a bonus/malus added to the learned action values (Fig 2A): For Win cues, a bonus is added to the value of Go actions, and for Avoid cues, a malus is subtracted from the value of Go actions [8]. This bias is present at the onset of the task and constant throughout learning. Since action values and the bias jointly

determine which action is selected, and since action values will increasingly separate with learning over the course of the task, the role of this response bias will, in net, decrease over the time [9].

The second bias, called **learning bias**, captures two tendencies well described in the literature: people ascribe positive outcomes to their own actions (which in other contexts can lead to "illusions of control" [33,34], but tend to ignore negative outcomes after having remained passive (a form of "omission bias" [35,36]; Fig 2B). These tendencies are captured by a boost in the learning rate for rewarded Go actions and a decrement in the learning rate for punished NoGo actions [9]. The impact of this bias on behavior only develops over time, where the action values become increasingly distorted relative to unbiased learning.

The third bias, called **persistence bias**, captures the fact that the cue valence (Win or Avoid cue) can act as a reinforcer (Fig 2C) [10]: For Win cues, rewards signal positive feedback, and neutral outcomes signal negative feedback. However, the Win cue itself can erroneously be interpreted as a feedback signal and reinforce behavior irrespective of the eventual outcome. Thus, while a neutral outcome for a Win cue signals negative feedback, which suggests switching behavior, the cue valence itself might be taken for positive feedback, which usually translates into a response repetition. Vice versa, for Avoid cues, neutral outcomes signal positive feedback and punishments signal negative feedback. The Avoid cue itself can erroneously be interpreted as negative feedback. Thus, while a neutral outcome for an Avoid cue signals positive feedback, which suggests repeating behavior, the Avoid cue itself can be taken for negative feedback, which usually translates into a response switch. We have previously observed that this bias can dominate repeat/switch behavior after neutral outcomes, resulting in the paradoxical observation of more response repetition after negative (i.e., no reward) feedback for Win cues than positive (i.e., no punishment) feedback to Avoid cues [10]. Such a bias should be invisible at the beginning of the task and only arise with learning, leading to increasingly distorted learning curves over time.

We tested the presence of these three biases in behavior using a series of seven reinforcement learning models. The base model (M1) was a simple Q-learning model that learned the values of Go and NoGo actions for each cue. To generate choices, action values were turned into action probabilities via a softmax transform. Action values were subsequently updated based on the obtained outcomes via reward prediction errors, which were scaled by a free learning rate parameter $\varepsilon$ and then added to the old action values. In line with previous implementations [8–10], objective outcomes were scaled by a free feedback sensitivity parameter $\rho$. High values on this parameter lead to learned action values developing further apart and thus more deterministic choices, similar to a high inverse temperature parameter used in alternative models. We extended this model with a constant Go bias parameter b to capture overall tendencies towards Go or NoGo responses (M2), a response bias parameter $\pi$ that added a bonus/malus to the action value of Go depending on the valence of a cue (M3), and finally, a learning bias parameter $\kappa$ capturing increased learning rates for rewarded Go responses and decreased learning rates for punished NoGo responses (M4). M5 combined all three parameters and fitted the data better than any of the simpler models, suggesting that both a response bias and learning bias were present in the data. Finally, we extended M5 by adding a persistence parameter $\varphi_{INT}$ that captured overall tendencies to repeat versus switch actions (M6), and a persistence bias $\varphi_{DIFF}$ that captured the higher tendency to repeat actions for Win cues than for Avoid cues (M7).

We compared these seven models using Bayesian model selection [37]. In line with our previous work [10], M7, implementing all three Pavlovian biases, was the best-fitting model according to the log-model evidence (Fig 2G), the model frequency (M7: 61%; M6: 14%; M5: 25%), and crucially, according to the protected exceedance probability (M7: 98%; M5: 2%; Fig 2K). We performed posterior-predictive checks by simulating the action probabilities based on the fitted parameters combined with participants' actions and outcomes (one-step ahead predictions), which captured the empirical data patterns very well (Fig 2H–2J). Simpler models failed to capture core features of the empirical data patterns. Model recovery showed that all models were distinguishable from each other, and parameter recovery on M7 showed that all parameters were well identifiable (S6 and S7 Figs). We also explored alternative models in which cue valence did not affect

persistence directly, but biased the outcomes in the prediction error updating equation. Such models predicted, qualitatively, a very similar pattern as M7, but showed slightly inferior fit (S5 Fig). In sum, these results confirmed the presence of all three Pavlovian biases in participants' behavior in the sham sessions.

We again tested for effects of age, gender, session ID, and session order on parameter values (using estimates from all three sonication conditions). Male participants showed overall significantly higher values for the reward sensitivity $\rho$ ($\chi^2(1)$ = 6.037, $p$ = .014), in line with more deterministic behavior and higher accuracy. There were no significant effects of age on parameter values. Across session IDs, values of both $\rho$ ($\chi^2(1)$ = 10.336, $p$ = .006) and $\varepsilon$ ($\chi^2(1)$ = 6.267, $p$ = .044) significantly increased, in line with increasing accuracy across sessions. In consequence, values of $\pi$ significantly decreased across sessions ($\chi^2(1)$ = 8.423, $p$ = .015), which is expected since a lower number of errors will lead to a smaller Pavlovian response bias. Finally, there was no effects of session order on any parameter value.

## Sonication effects on Go/NoGo choices

We next analyzed the effect of TUS on Pavlovian biases in responding and learning. We again first used regression models based on participants' Go/NoGo responses and their propensity to repeat/switch responses, and then subsequently contrasted parameters obtained from the best-fitting reinforcement learning model across all sonication conditions.

We used a mixed-effects logistic regression model to test for effects of TUS on (biased) responding, i.e., their Go/NoGo responses. This model included the required action, cue valence, the TUS sonication condition, as well as the block half (first versus second half of a block) to account for the possibility that TUS might affect learning processes, which might only become visible in the second half of each block. Beyond the main effects of required action and cue valence and their interaction, this model yielded a significant 3-way interaction between required action, cue valence, and sonication condition, $\chi^2(2)$ = 14.277, $p$ < .001, a significant 3-way interaction between cue valence, sonication condition, and block half, $\chi^2(2)$ = 9.191, $p$ = .010, and a significant 4-way interaction between required action, cue valence, sonication condition, and block half, $\chi^2(2)$ = 21.548, $p$ < .001 (Fig 3A–3C). See S3 Table for results when analyzing each cue condition (Go-to-Win, Go-to-Avoid, NoGo-to-Win, NoGo-to-Avoid) separately. To better understand these effects, we followed them up with the regression analyses of response repetitions versus switches.

## Sonication effects on response repetitions/switches

Analyses of Go/NoGo choices suggested effects of TUS on learning (i.e., changes in Go/NoGo responses over time). To more directly investigate learning, i.e., the impact of outcomes on subsequent choices, we analyzed participants' repeat/switch choices as a function of outcomes and responses on the current trial. We focused on the signatures of biased learning.

First, we contrasted sonication conditions selectively on those trials on which biased learning should occur, namely trials with rewarded Go responses or with punished NoGo responses. If the learning bias became stronger, participants would repeat rewarded Go responses more and punished NoGo responses less, yielding a stronger difference between both conditions (with no change in any other condition). Vice versa, if the bias became weaker, the difference between both conditions would decrease. In a mixed-effects logistic regression model fit to the response repetitions versus switches, we observed a significant main effect of outcome (reward versus punishment) qualified by a significant outcome × sonication interaction, $\chi^2(2)$ = 6.162, $p$ = .046: participants more often repeated rewarded Go responses than punished NoGo responses and this tendency was significantly attenuated after TUS-aIns relative to sham, $b$ = −0.150, 95%-CI [−0.293, −0.006], $\chi^2(1)$ = 4.197, $p$ = .040, and not after TUS-dACC compared to sham, $b$ = 0.001, 95%-CI [−0.058, 0.249], $\chi^2(1)$ = 0.001, $p$ = .984 (Fig 3C). These results suggest that TUS-aIns, but not TUS-dACC, reduced the learning bias.

Second, we tested whether sonications changed the effect of cue valence on response repetitions versus switches, reflecting a bias in persistence. Participants were more likely to repeat responses for Win than Avoid cues (main effect of valence: $b$ = 0.428, 95%-CI [0.342, 0.514], $\chi^2(1)$ = 94.620, $p$ < .001], which was qualified by a cue valence × sonication

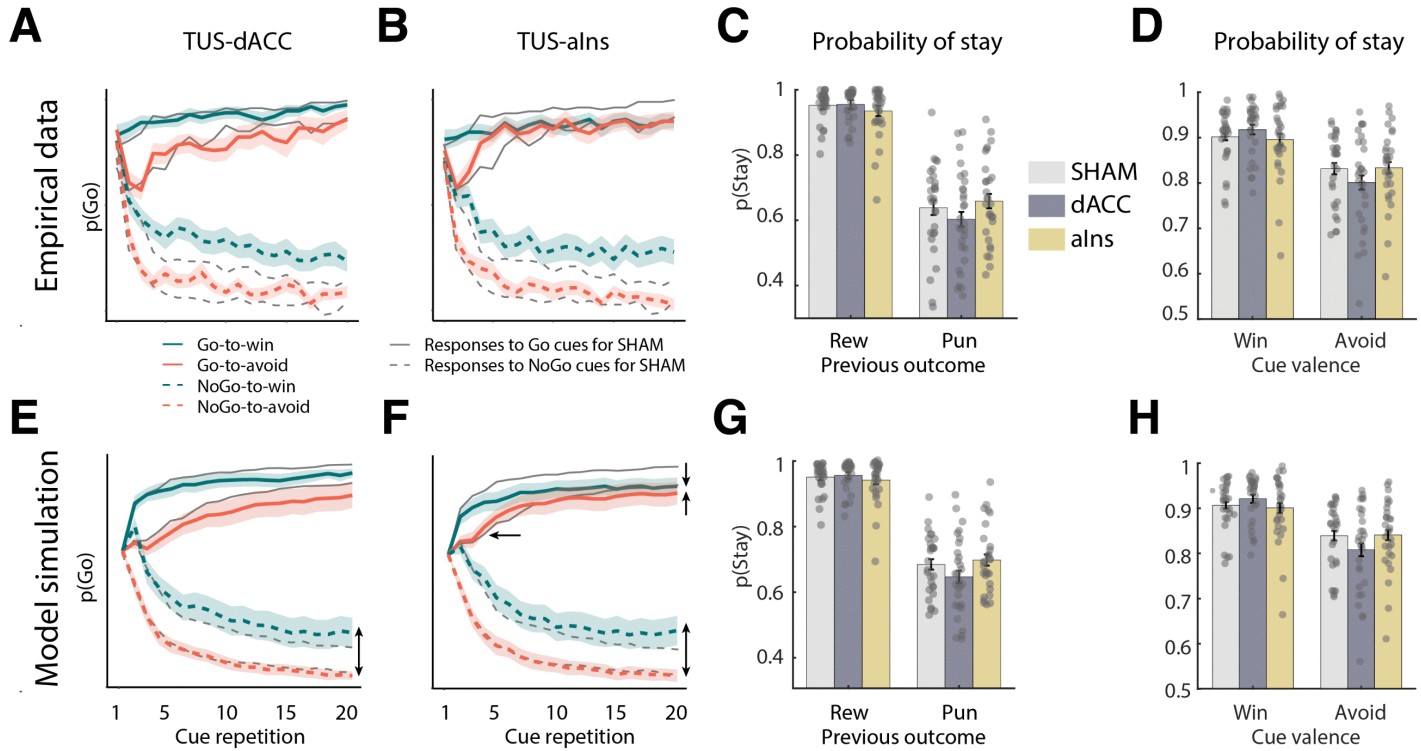

**Fig 3. Empirical and model-based effects of TUS on Go responding and response repetition biases. A, B Empirical Probability of Go.** Trial-by-trial proportion of Go responses across sonication conditions (TUS-dACC, and TUS-aIns, gray lines represent the sham condition). TUS-dACC decreased Go responses in the Go-to-Win condition but increased them in the NoGo-to-Win condition. TUS-aIns decreased Go responses in the Go-to-Win condition but increased them in the Go-to-Avoid and NoGo-to-Win conditions. Coloured lines display the active sonication condition. **C, D Probability of Stay.** TUS effects on response repetitions based on outcomes (rewarded Go vs. punished NoGo) and cue valence (Win vs. Avoid). TUS-aIns (but not TUS-dACC) attenuated the tendency to repeat rewarded Go responses, but increased the tendency to repeat punished NoGo responses compared to sham. TUS-dACC (but not TUS-aIns) increased the cue valence effect on response repetitions, with stronger persistence for Win cues relative to Avoid cues. **E–H** Posterior predictive checks reproduced the empirical data patterns for the probability of Going and staying. The data underlying this figure can be found in S3 Data.

interaction, $\chi^2(2) = 7.617$, $p = .022$: the cue valence effect was significantly stronger after TUS-dACC compared to sham, $b = 0.113$, 95%-CI [0.031, 0.194], $\chi^2(1) = 7.368$, $p = .007$, with no difference between TUS-aIns and sham, $b = 0.006$, 95%-CI [−0.063, 0.074], $\chi^2(1) = 0.025$, $p = .873$ (Fig 3D). These results suggest that TUS-dACC, but not TUS-aIns, affects the persistence bias. Next, we directly confirmed these observations by fitting these parameters using computational reinforcement learning models and comparing the fitted parameters across TUS sonication conditions.

## Effects of TUS on computational modeling parameters

We fit the best-fitting model from the sham session, M7, comprising all three Pavlovian biases, namely a response bias π, a learning bias κ, and a persistence bias $\varphi_{DIFF}$, to the other two sonication conditions. Model comparisons suggested that, in the TUS-dACC sessions, M7 was again the best-fitting model according to both the model frequency (M7: 89%; M6: 11%) and the protected exceedance probability (M7: 100%). In the TUS-aIns sessions, a slightly different outcome occurred, with M6 being the best-fitting model according to both the model frequency (M6: 68%; M7: 32%) and protected exceedance probability (M6: 97%; M7: 3%). Given that M6 is a nested version of M7 (i.e., M7 with the $\varphi_{DIFF}$ parameter fixed to zero is equivalent to M6), we proceeded with comparing the parameter values in M7 across all three sonication conditions.

We focused on the three parameters reflecting the three different bias parameters: the response bias π, the learning bias κ (Fig 4A), and the persistence bias $\varphi_{DIFF}$ (Fig 4D). We compared the parameter values between all three conditions with a one-way repeated-measures ANOVA, following up any significant effects with paired *t*-tests. To correct for multiple comparisons (seven tests for seven parameters), we applied an adjusted alpha-level of .05/ 7 = .007. There was no significant difference in the response bias π, $F(2, 56) = 1.719$, $p = .189$, generalized $\eta^2 = 0.018$. However, there was a significant difference in the learning bias κ, $F(2, 56) = 7.680$, $p = .001$, $\eta^2 = 0.134$: this bias was significantly attenuated after TUS-aIns compared to sham, $t(28) = 4.381$, $p < .001$, Hedges' $g = -0.791$, bootstrapped 95%-CI [−1.248, −0.460], but not after TUS-dACC compared to sham, $t(28) = 0.489$, $p = .629$, $g = -0.088$, 95%-CI [−0.494, 0.264] (Fig 4B). This observation is in line with the regression results showing an attenuated difference in repetitions of rewarded Go versus punished NoGo responses after TUS-aIns.

Furthermore, there was a significant difference in the persistence bias $\varphi_{DIFF}$, $F(2, 56) = 9.836$, $p < .001$, $\eta^2 = 0.157$: this bias was significantly stronger after TUS-dACC compared to sham, $t(28) = 4.028$, $p < .001$, $g = 0.728$, 95%-CI [0.419, 1.146], but not after TUS-aIns compared to sham, $t(28) = 1.058$, $p = .299$, $g = 0.191$, 95%-CI [−0.170, 0.547] (Fig 4F). This effect was in line with the increased cue valence effect on response repetitions versus switches after TUS-dACC. In sum, TUS-aIns attenuated the learning bias, while TUS-dACC increased the persistence bias.

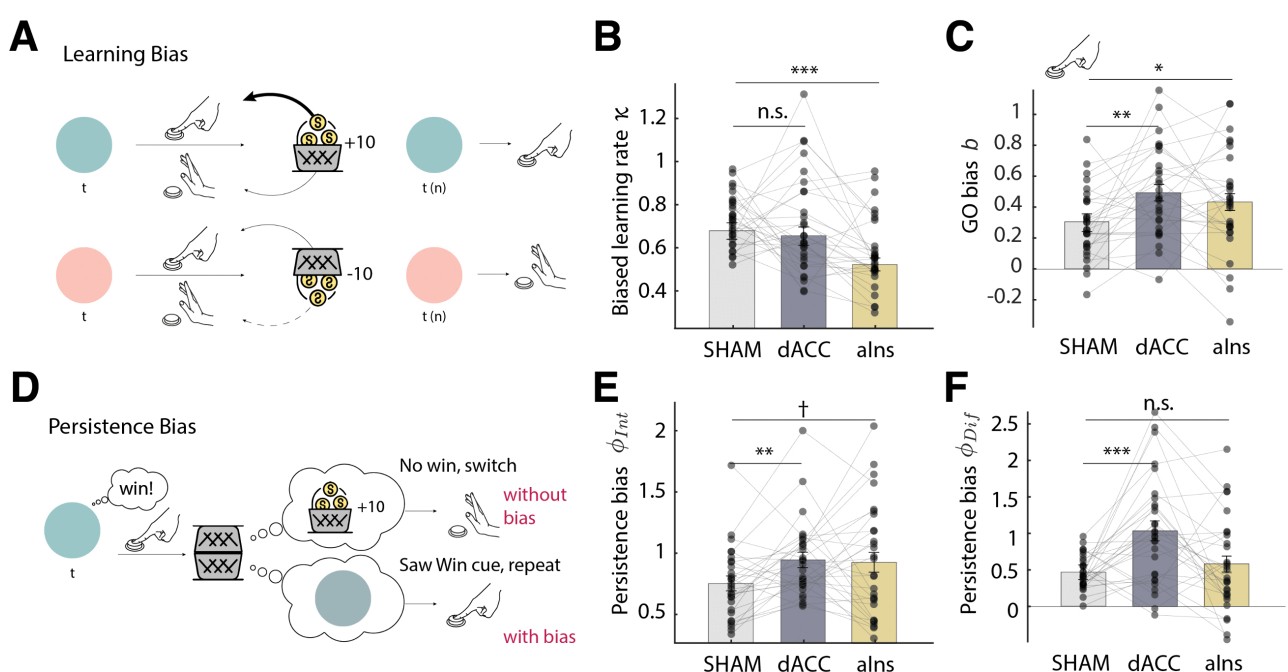

**Fig 4. Effects of TUS on instrumental learning biases, as reflected by computational model parameters. A.** Learning bias illustration: Learning is stronger for rewarded Go actions, and weaker for punished NoGo actions. **B.** Learning bias (κ): TUS-aIns (but not TUS-dACC) significantly diminished the learning bias, i.e., the tendency to attribute rewards to actions and the reluctance to attribute punishments to inactions. **C.** Go bias (b): both sonication conditions significantly increased participants' overall tendency for Go responses regardless of the cue valence. **D.** Persistence bias illustration: participants show an overall higher tendency to repeat actions for Win cues than for Avoid cues. For Win cues, this implies that they sometimes ignore neutral outcomes (negative feedback, which suggests to switch the response) and instead take the Win cue itself as positive feedback (which suggests to repeat the response). **E.** Persistence parameter ($\varphi_{INT}$): both sonication conditions increased participants' overall tendency to repeat previous responses. **F.** Persistence bias ($\varphi_{DIFF}$): TUS-dACC (but not TUS-aIns) significantly increased the persistence bias, leading to a stronger tendency to repeat responses relatively more for Win cues than for Avoid cues. Asterisks indicate significance levels: ***$p < 0.001$; **$p < 0.01$; *$p < 0.05$; † $p < 0.10$. The data underlying this figure can be found in S4 Data.

Beyond the bias parameters of interest, both TUS conditions also unspecifically increased the Go bias parameter, $F(2, 56) = 4.958$, $p = .010$, $\eta^2 = 0.071$ (Fig 4C), and the persistence parameter $\varphi_{INT}$, $F(2, 56) = 3.517$, $p = .036$, $\eta^2 = 0.055$ (Fig 4E), though neither of these changes was significant when corrected for multiple comparisons. We report follow-up $t$-tests for completeness. Relative to sham, the overall Go bias $b$ was significantly increased after both TUS-dACC, $t(28) = 3.148$, $p = .004$, $g = 0.569$, 95%-CI [0.228, 0.979], and TUS-aIns, $t(28) = 2.130$, $p = .042$, $g = 0.385$, 95%-CI [0.039, 0.774]. Similarly, the overall persistence parameter $\varphi_{INT}$ was significantly increased after TUS-dACC, $t(28) = 3.070$, $p = .005$, $g = 0.555$, 95%-CI [0.211, 1.000], and marginally significantly increased after TUS-aIns, $t(28) = 1.972$, $p = .059$, $g = 0.356$, 95%-CI [0.026, 0.683] (not significant when correcting for multiple comparisons). Notably, stimulation effects on $\kappa$ and $\varphi_{DIFF}$ were still significant when controlling for b and $\varphi_{INT}$ using mixed-effects linear regression with a random intercept per participant overall effect of sonication on $\kappa$: $\chi^2(2) = 18.591$, $p < .001$; difference TUS-aIns–sham in $\kappa$: $\beta = 0.546$, 95%-CI [0.313, 0.780], $\chi^2(1) = 21.001$, $p < .001$; overall effect of sonication on $\varphi_{DIFF}$: $\chi^2(2) = 13.378$, $p < .001$; difference TUS-dACC–sham in $\varphi_{DIFF}$: $\beta = 0.374$, 95%-CI [0.126, 0.622], $\chi^2(1) = 8.758$, $p = .003$), suggesting that region-specific effects occurred independently of these unspecific effects. Furthermore, stimulation effects remained significant when controlling for age, gender, session ID, or session order, and did not interact with any of these variables. There was no significant difference in the feedback sensitivity parameter $\rho$, $F(2, 56) = 0.928$, $p = .401$, $\eta^2 = 0.015$, or the learning rate $\varepsilon$, $F(2, 56) = 1.047$, $p = .358$, $\eta^2 = 0.018$.

We tested whether the effect of TUS on behavior could be predicted by the dose of the ultrasound stimulation. For this purpose, we correlated the changes in parameter values (active session – sham) with the estimated dose (pressure in kPa) based on acoustic simulations of the acoustic beam trajectory. These simulations were based on the achieved target and transducer coordinates recorded during the stimulation sessions. After controlling for multiple comparisons (Bonferroni correction for eight tests, adjusted alpha-level of $.05/8 = .006$), we did not observe any significant relationship between pressures and TUS effects on parameters (S11 Fig).

Taken together, the effects of dACC and aIns sonication were best captured by a decrease in the learning bias after TUS-aIns, an increase in the persistence bias after TUS-dACC, and increases in overall Go biases and persistence after both types of sonications. These effects were well captured by posterior predictive checks: Simulated learning curves closely followed the empirical learning curves (Fig 3E and 3F). These observations also match the regression results of response repetitions/switches, with an attenuated difference between rewarded Go and punished NoGo responses after TUS-aIns (Fig 3G) and a stronger cue valence effect after TUS-dACC (Fig 3H). In sum, in this study, we found a functional differentiation between the effects of TUS-aIns and TUS-dACC, with both altering different Pavlovian biases.

## Discussion

This study investigated the effects of TUS to the dACC and aIns on Pavlovian biases using a Motivational Go/NoGo task. Participants showed three forms of Pavlovian biases: a response bias, learning bias, and persistence bias [9,10]. TUS revealed a functional differentiation, with dACC and aIns affecting distinct learning processes: TUS-aIns attenuated a learning bias, reducing the tendency to attribute rewards to actions and reluctance in attributing punishments to inactions, while TUS-dACC increased a persistence bias, strengthening the tendency to use cue valence as a reinforcer signal and persist after positive (compared to negative) cues. Both stimulation conditions also affected Go responses and response persistence in an unspecific manner, though these effects were weak and not significant when correcting for multiple comparisons. Given that dACC and aIns are part of the same resting-state network, the saliency network [38,39], and regularly co-activate in cognitive tasks [40,41], one might expect that stimulating these regions leads to largely similar effects. While this is the case for such unspecific effects, each region also showed a unique effect, suggesting that distinct regions of the saliency network might have distinct roles in learning and response adjustment. Overall, it is likely that both learning processes arise from the interplay of a network of regions, and our results suggest a relative predominance of the dACC and aIns in these processes, respectively. Our study contributes to understanding their different roles in learning from

feedback: While the dACC appear important in selecting which environmental signals to use as feedback, the aIns seems important in credit assignment whether these signals were caused by an agent's own actions.

We set out this study under the hypothesis that perturbing neural activity in dACC and aIns, which are part of an early "reset" network that rapidly responds to negative/salient stimuli [16], could alter the strength of Pavlovian biases. In particular, several previous studies have observed a selective role of dACC and aIns in choices to avoid losses [42,43], learning from negative feedback [44,45], and adjusting behavior [46,47]. However, in our results, TUS did not selectively affect responses to negative (Avoid) cues. Specifically, TUS did not change the cue valence effect (i.e., response bias), the required action effect (i.e., overall accuracy), or their interaction (i.e., a change in accuracy for only Avoid cues). Similarly, in our computational models, (baseline) learning rates and the response bias were unaffected by TUS. Instead, the effects of TUS only became apparent over time, indicative of changes in learning processes, and were spread across all four cue conditions. These findings reflect more intricate changes in learning that were well captured by a computational model [10]. This finding might not be surprising given a broad literature implicating dACC and aIns in learning from feedback and adjusting action policies [20,48,49].

We found that stimulating the aIns with TUS led to a reduction in a learning bias: participants showed a reduced tendency (i.e., lower learning rate) to ascribe rewards to their own actions, as well as a reduced reluctance to ignore punishments of their own inactions. Such biases have been interpreted as global "priors" on which action-outcome relationships likely hold in an organism's environment [9,10]. Using such priors, individuals can trade-off noisy outcomes of single actions against prior hypotheses on what is generally the best response strategy. Such biases will facilitate the acquisition of actions that are conducive to rewards but slow down the acquisition of actions in contexts in which punishments are involved. While such biases will usually be adaptive and lead to more robust learning, the tendency to attribute rewards to own actions can lead to the false inference of "illusionary control" over outcomes that are in fact random [50,51]. Similarly, these biases could explain why humans judge actions that lead to negative outcomes more harshly than omissions leading to such outcomes ("omission bias") [52], a phenomenon cited to explain some humans' reluctance to vaccinate [35,53].

We have previously found that neural signals associated with such biased learning are first visible in the blood-oxygen-level-dependent (BOLD) signal from cortical regions including the perigenual anterior cingulate cortex (pgACC) and posterior cingulate cortex (PCC), prior to subcortical regions (striatum) [10]. Previous research has indeed suggested cortical regions, most prominently the dACC, to control the speed of learning (learning rates) in subcortical regions [48,49,54]. Although related bias phenomena, such as auto-shaping and negative self-maintenance, have been shown in animals such as rodents or pigeons [33,55], we have previously speculated that a prefrontal, cortical basis of this bias in humans might hint at a more recent innovation in primates that is responsible for this particular learning bias.

Given the previous literature on the role of the dACC in scaling learning rates [49,56,57], we would have a priori expected that TUS dACC should have affected the learning bias, and a selective reduction under TUS-aIns might at first appear puzzling. The aIns has previously been found to encode the uncertainty/risk under which a choice is made [58–60] as well as the average reward rate in an environment [25], which are key environmental variables that should scale learning rates according to normative models [61–63]. Overall, these findings suggest that aIns plays a crucial role in using higher-order beliefs about the environment to decide how much to update lower-order beliefs based on reward/punishment feedback. This supports a role of the aIns in scaling learning rates, which can give rise to learning biases.

An alternative possibility is that the TUS-aIns effect reducing biased credit assignment arises from its role in social cognition and self-other attribution. Increased aIns activity has been shown both when an agent is directly affected by an outcome (e.g., pain, unpleasant taste) and when they observe such outcomes in another individual [24,64,65]. Hence, the computational challenge arises for the aIns to tell apart which outcomes were caused by oneself and which were merely observed in others [24]. TUS-aIns might blur this boundary, leading to changes in credit assignment as observed in the current study. Some studies have speculated that this role is subserved by von Economo and Fork neurons, which, within

the primate brain, occur almost exclusively in the aIns [66,67] and have only been observed in animal species that live in large social groups [68]. This speculative interpretation suggests a dedicated role of the aIns in credit assignment, following an evolutionary route that might be special in primates. Investigating the causal contribution of such deep cortical circuits to learning from feedback might greatly benefit from the use of non-invasive brain stimulation tools such as TUS. Taken together, our results suggest that aIns plays an important role in credit assignment and deciding whether environmental feedback signals were caused by personal actions.

In contrast to TUS-aIns, TUS-dACC selectively affected the persistence bias, with an increased tendency to repeat actions for Win cues than Avoid cues, regardless of the actions and outcomes they had been followed by. This bias reflects participants' tendency to use the cue valence instead of the outcome valence as a feedback signal [10]. Knowledge of the cue valence is important for disambiguating neutral outcomes, which can only be used as a teaching signal when contrasted against the counterfactual that could otherwise have been obtained (i.e., a neutral outcome signaling a missed reward is negative feedback; a neutral outcome signaling a missed punishment is positive feedback). However, human participants show limited ability to tell apart counterfactual and real outcomes, sometimes taking a neutral outcome obtained instead of a reward as a positive outcome, and a neutral outcome obtained instead of a punishment as a negative outcome [10], a tendency exacerbated by TUS-dACC. Previous studies have implicated a causal role of the dACC in learning from counterfactual feedback [20] and deciding when it is worth switching an action policy [23,69,70]. When perturbed with TUS, the dACC might mediate a confusion between real and counterfactual feedback, leading to a decreased ability to switch behavior after a missed reward, which gives rise to larger persistence bias. In sum, our results suggest that the dACC is important in selecting which environmental signals to use as reinforcer to adjust behavior.

Interestingly, we found similar unspecific effects of TUS-dACC and TUS-aIns on the overall propensity to make Go responses and to repeat previous responses (though not significant when correcting for multiple comparisons), suggesting that sonication of both regions had similar effects in some respects. This observation tentatively suggests that the directionality of the effect (i.e., excitation or inhibition) was the same for both stimulation sites. Previous studies that have used the same stimulation protocol on the posterior cingulate cortex and basolateral amygdala have used magnetic resonance spectroscopy (MRS) to directly measure excitation-inhibition (E/I)-balance in the stimulated regions [22,71]. These studies found an inhibitory effect, suggesting that TUS perturbs "normal" activity these regions. Under such a perturbatory influence, one might indeed expect a decrease of the learning bias $\kappa$ after TUS-aIns, and a failure to inhibit the misleading influence of cue valence after TUS-dACC However, in this study, we did not measure MRS and thus cannot directly test the directionality of the effect on neural activity. Notably, one of the studies using MRS mentioned above [71] found an excitatory effect of the same protocol in the mid-insula, tentatively suggesting that the same stimulation protocol might have different effects in different regions.

More broadly, it remains an open question whether a given ultrasound protocol induces uniform neural effects across different brain regions. Emerging evidence suggests that this is unlikely to be the case. Differences in local cytoarchitecture, cell-type composition, and baseline functional state may strongly constrain how ultrasound interacts with neural tissue [72,73]. It is possible that region-specific morphology, circuits and functions shape the functional consequences of ultrasound, even when stimulation parameters are held constant. From this perspective, the effects observed in dACC and aIns may reflect a shared perturbation of task-relevant computations rather than identical cellular mechanisms, highlighting the importance of considering regional and circuit-level context when interpreting the neural impact of TUS. Our findings also highlight the promise of TUS as a neuromodulation tool in cognitive neuroscience. Compared to other methods, such as TMS or transcranial direct current stimulation (tDCS), TUS allows for deeper, more focal targeting of subcortical and midline/lateralized deep cortical structures, making it particularly suited to probing the causal roles of regions like the aIns and dACC. As such, this method opens new possibilities for investigating the neural mechanisms of learning and decision-making. Furthermore, it might open new avenues in treating psychiatric disorders. In particular, in the context of the Motivational Go/NoGo Task, little is known about the underlying learning biases, but they show obvious relationships

with psychiatric symptoms: credit assignment processes are frequently found altered in schizophrenia, leading to an underestimation of one's own and overestimation of others' agency [74,75]. Similar, depression is characterized by a lack of sense of agency/self-efficacy [76,77]. In contrast, overly rigid behavior and perseverative processes are often observed in obsessive-compulsive disorder (OCD), where such symptoms might arise from altered feedback processing or selection of feedback signals [78,79]. Such links would have to be corroborated and the efficacy of TUS in treating such symptoms tested in further studies.

In summary, our study reveals a functional differentiation in the effects of TUS on learning biases, with TUS-aIns selectively attenuating the learning bias and TUS-dACC increasing the persistence bias. Hence, while dACC and aIns often co-activate in cognitive roles, they appear to have distinct roles in learning from feedback: while TUS to the aIns decreased people's tendency to overly take credit for rewards following action and to ignore punishments following inaction, TUS to dACC increased participants' tendency to take the cue valence as a reinforcer signal. These findings advance our understanding of the differential roles of deep cortical regions in instrumental learning and highlight the potential of TUS as a tool for modulating cognitive and affective processes. Our findings challenge the traditional view that learning biases are exclusively driven by subcortical regions associated with rigid, habit-like behaviors. Instead, these results emphasize the role of frontal inputs, which contribute to counterfactual reasoning and behavioral flexibility [20,49]. The involvement of these two areas in biased learning suggests that these biases may not be rigid constraints, but rather adaptive priors that guide how individuals integrate past experiences. By balancing prior beliefs about action-outcome associations with flexible learning from rewards and punishments, these biases may enhance decision-making in dynamic environments [4,10].

## Methods

### Participants

Thirty-four healthy participants (18 females, 16 males) aged 19–59 years (mean age = 29.73, SD = 11.16) took part in this study. One participant was excluded because they did not understand the task and three because of missing sessions. Twenty-nine participants (16 females, 13 males), aged 19–59 years (mean age = 30.24, SD = 11.63) were included in the analysis. Participants had normal or corrected-to-normal vision. All participants provided written informed consent following a full explanation of the study procedures. Ethical approval was granted by the University of Plymouth Faculty of Health Staff Research Ethics and Integrity Committee (reference ID: 3394; date: 28/06/22). The study was conducted in accordance with the principles expressed in the Declaration of Helsinki.

Prior to participation, all participants completed a structured procedure to assess eligibility for MRI and TUS. Exclusion criteria covered standard MRI contraindications and safety restrictions, pregnancy, age below 18 years, a history of major neurological conditions linked to increased seizure risk, relevant implanted or metallic devices, and current medication or substance deemed incompatible with the study safety criteria. Participants were additionally instructed to abstain from alcohol for at least 24 hours before stimulation. Adverse events were assessed after each TUS session and again 12–48 hours later using a structured symptom questionnaire in which participants rated symptom severity and their perceived relationship to stimulation. The full versions of the screening and post-TUS questionnaires are provided in the Supporting information. No participant reported a current neurological or psychiatric diagnosis or any use of psychoactive medication.

Participants were compensated an average of £85, which included a performance bonus of up to £10 based on points collected over three sessions in the Motivational Go/NoGo task. Travel expenses were reimbursed up to £5 per session. All study sessions took place at the Brain Research & Imaging Centre (BRIC) in Plymouth, UK.

### Study design

The study design is summarized in Fig 1A. Participants began with an MRI-only session, which included a series of scans, including a T1-weighted MRI scan. This imaging data set was used to create a personalized head model for neuronavigation and acoustic and thermal simulations, enabling precise planning of the TUS target and transducer placement for the

upcoming TUS sessions. In the first session, participants completed a practice phase to become familiar with the Motivational Go/NoGo learning task.

For the TUS sessions, participants received a randomized sequence of TUS conditions. Each session involved an 80-s TUS application with three different conditions: active stimulation of the bilateral anterior insula (TUS-aIns), active stimulation targeting the dorsal anterior cingulate cortex (TUS-dACC), and a sham condition. In the sham condition, the transducer was positioned over the temporal region bilaterally, similar to the TUS-aIns setup, but no ultrasound was delivered. To maintain a consistent procedure, both sonication and sham conditions were always applied first to the left hemisphere, followed by the right. Directly after the end of the stimulation procedure (maximal latency of 10 min), participants started the Motivational Go/NoGo Task.

## MRI data acquisition

For TUS planning, including neuronavigation and acoustic simulations, MRI scans were acquired using a Siemens MAGNETOM Prisma 3T scanner (syngo MR V11E, Siemens Healthineers, Erlangen, Germany) equipped with a 64-channel head coil. The scanning protocol for this study included a T1-weighted MPRAGE sequence in the sagittal plane (TR = 2100 ms, TE = 2.26 ms, inversion time = 900 ms, flip angle = 8°, GRAPPA acceleration factor = 2, matrix size = 256 × 256, 176 slices, voxel size = 1 × 1 × 1 mm$^3$).

## TUS protocol and procedure

Targeting of the bilateral aIns and dACC was guided by personalized anatomical landmarks using each participant's T1-weighted MRI scans during the planning of target and transducer placement (see S1 Table). Acoustic field simulations, generated from pseudo-CT skull models derived from participants' T1-weighted images using a convolutional neural network (CNN) [29], ensured precise targeting and safety by keeping both on-target and off-target pressure within acceptable ranges.

A bespoke CTX-500 NeuroFUS TPO system (Brainbox, Cardiff, UK) with a four-element annular transducer (diameter = 64 mm, central frequency = 500 kHz, and steering range between 27.3 and 82.6 mm) (S1 Fig) was used to deliver the TUS protocol. The TUS protocol consisted of repetitive TUS applied using a 5 Hz-patterned protocol with a 10% duty cycle, lasting 80 seconds and delivering a total of 400 pulses (pulse duration of 20 ms and pulse interval of 200 ms). The initial spatial-peak pulse-average intensity (ISPPA) in free-field conditions was set at 50 W/cm² for all participants. Free-field pressures were calibrated in a water tank using an ONDA capsule hydrophone (HGL-series Golden Lipstick) mounted in a low-cost hydrophone tank adapted from [80] converting voltage from the stable portion of the waveform at focus to pressure using the hydrophone's calibrated sensitivity to compute peak rarefactional pressure and ISPPA. No attenuation correction was applied for the free-field water calibration beyond the intrinsic frequency-dependent sensitivity correction of the hydrophone (water assumed lossless at 500 kHz).

We took care to remain within the guidelines for human ultrasound exposure as defined by ITRUSST [30]. The results of our acoustic simulations predominantly indicated a maximum skull temperature rise below 2 °C (S1 Table). In cases where this threshold was exceeded, we calculated the CEM43, a metric reflecting both duration and intensity of heating relative to 43 °C, the critical threshold for thermal cell damage. We ensured that CEM43 values remained well below 0 [30]. In our study, CEM43 was always below 0.1. The transcranial Mechanical Index (MItc) was computed as $MItc = P\_neg,tc/\sqrt{f}$, where $P\_neg,tc$ is the simulated peak rarefactional (negative) pressure after skull transmission (MPa) and f is the ultrasound fundamental frequency in MHz (0.5 MHz). $P\_neg,tc$ values were extracted as voxelwise maxima within a 3 mm-radius spherical ROI centered on the stimulation target, likely to include local maxima.

To improve ultrasound transmission, a layer of ultrasound gel (Aquasonic 100, Parker Laboratories) was applied at the transducer placement site, with a 2 cm gel pad (Aquaflex, Parker Laboratories) positioned between the transducer and the participant's head. Participants' heads remained unshaved, and any air bubbles were carefully eliminated by smoothing

and combing the hair. Neuronavigation was performed using Brainsight v2.5 (Rogue Research, Montréal, Québec, Canada) with T1-weighted anatomical MR images. During each session, focal depth measurements obtained from Brainsight were entered into the NeuroFUS TPO system (Brainbox, Cardiff, UK) before stimulation, and the trajectory was sampled to support confirmatory acoustic simulations following each session.

The dACC, was defined anatomically as the region anterior to the anterior commissure, posterior to the genu of the corpus callosum, bounded ventrally by the callosal sulcus. We aimed to stimulate just ventrally of the cingulate sulcus such as to engage the area labeled aMCC by Palomero-Gallagher and colleagues [81], with a maximum intensity on region 24c'v. The aIns target was localized in the anterior short gyrus located deep within the Sylvian fissure and bordering the frontal operculum.

## Blinding procedure

Sham TUS was administered identically to active TUS, except that no actual stimulation was applied. Instead, participants listened to a sound mimicking the transducer's pulse repetition frequency through bone conduction headphones, positioned approximately 2 cm posterior and superior to the temples. This sound was only played during the Sham condition to ensure blinding, while headphones were worn in both active and Sham sessions. To create this signal, we generated a waveform matched to the ultrasound protocol in terms of pulse repetition frequency and pulse duration, and we adjusted the sampling rate to approximate the perceptual pitch of the TUS-related sound. Six lab members who had previously experienced the full TUS protocol evaluated several candidate waveforms and selected the one that most closely resembled the real stimulation. This calibration step had to be conducted in a separate group to avoid contaminating expectations in the actual study participants. During the experiment, the audio was played only during the sham condition.

## Task

The task consisted of four blocks, presented approximately 15, 25, 35, and 45 min after TUS (Fig 1B and 1C). Each block featured four cues (gems), with each trial requiring participants to determine, through trial and error, the correct action (Go or NoGo) and the potential outcome (reward or punishment) associated with each stimulus. The task design involved four different cues that varied by cue valence (Win versus Avoid) and required action (Go versus NoGo), counterbalanced across participants in three sessions of 320 trials each. Motivationally incongruent cues were included, where participants' natural action tendencies (e.g., to act for a reward) conflicted with the task requirements (e.g., to refrain from action when a reward was expected). For each cue, correct responses resulted in positive outcomes (80% chance of reward or avoiding punishment), whereas incorrect actions had only a 20% chance of yielding positive outcomes. The valence (Win or Avoid) of each cue was not explicitly signaled and had to be learned over time.

This probabilistic task was designed to capture learning and decision-making under uncertainty, similar to real-world scenarios where outcomes are not guaranteed. The structure of probabilistic feedback allowed us to observe participants' flexibility in adjusting strategies, sensitivity to feedback, and the cognitive and neural processes underlying decisions when faced with motivational conflicts, such as balancing exploration versus exploitation in uncertain environments.

## Stimuli display

The probabilistic motivational learning task was displayed on an LCD screen (1,920 × 1,080 resolution, 60 Hz refresh rate, 8-bit color depth, RGB, standard dynamic range (SDR)), positioned 0.5 m in front of participants. The experiment was programmed using the Presentation software (Neurobehavioral Systems, Berkeley, CA, USA) and run on a Windows-based computer. Participants responded using their preferred hand by pressing the SPACE bar on a keyboard.

## Post-TUS questionnaire

After each TUS session, as well as on the following day, participants were asked to report any adverse effects. They were presented with a list of symptoms and asked to rate the intensity to which they experienced each of them using a

4-point scale (absent, mild, moderate, severe) and whether they thought their experience was related to the stimulation on a 5-point scale (unrelated, unlikely, possible, probable, definite). Symptoms included those reported in [82], to which we added speech problems, vision problems, and muscle tightness of the face or arm. Respondents had the possibility of listing and rating up to two additional symptoms and to provide details for their experience.

For further insights, S1 Table present detailed information regarding the acoustic simulation parameters and the outcomes observed across participants. This includes data on mechanical indices, derated ISPPA values at the focal point, and the incidence of side effects associated with both aIns and dACC stimulation.

### Regression analyses

We fit mixed-effects logistic regression models using the lme4 package in R. We always included a random intercept per subject and all possible random slopes and random correlations, achieving a maximal random effects structure [83]. We used sum-to-zero coding for all categorical predictors. We obtained $p$-values from Wald chi-squared tests using Type-3 sums-of-squares via the Anova() function from the car package.

### Computational modeling and model comparison

We fit a series of increasingly complex computational reinforcement learning models to participants' choices, following previous studies [8–10]. All models are Q-learners that learn the expected value of performing an action for a given cue using reward prediction errors and select actions with higher expected value. On a given trial t, the base model **M1** computes choice probabilities for Go and NoGo actions (a) given a cue (s) using action weights (w; modified Q-values) turned into probabilities via a softmax transform:

$$p\left(a_t \mid s_t\right) = \frac{\exp(w\left(a_t, s_t\right))}{\sum_a \exp(w\left(a', s_t\right))}$$

(1)

For each action, the model obtains an outcome $r$ and computes a reward prediction error as the difference between the outcome $r$ and the expected value $Q(a, s)$. It then uses prediction errors to update $Q$-values via the delta learning rule:

$$Q_t\left(a_t, s_t\right) = Q_{t-1}\left(a_t, s_t\right) + \varepsilon(\rho r - Q_{t-1}\left(a_t, s_t\right))$$

(2)

Outcomes can be +1 (reward) or 0 (no reward) for Win cues, or 0 (no punishment) or −1 (punishment) for Avoid cues. In this learning rule, prediction errors are scaled by the learning rate $\varepsilon$, with large learning rates leading to faster updating (and a stronger recency bias in value estimates) and small learning rates leading to slower updating (and a more long-lasting impact of outcomes obtained far in the past). Furthermore, outcomes themselves are scaled with a participant-specific feedback sensitivity parameter $\rho$, which plays a similar role as an inverse temperature parameter: high values of **$\rho$** lead to $Q$-values for Go and NoGo actions developing further apart with learning and leading to more deterministic choices, while small values of $\rho$ lead to $Q$-values closer together and thus more stochastic choices. We initialized $Q$-values to the midpoint between the two possible outcomes of each cue (+0.5 for Win cues, −0.5 for Avoid cues) multiplied with participants' respective feedback sensitivity parameter such that starting $Q$-values are the midpoint of each participant's subjective value space. Given that the valence of each cue (Win/Avoid) was not instructed but had to be inferred from the attainment of valenced outcomes (rewards/punishments), participants could not interpret neutral outcomes as positive/negative feedback before having observed such a valence outcome. During this phase, $Q$-values were updated based on feedback, but "muted" in the action selection process (multiplied with zero when computing choice probabilities), assuming that participants could retrospectively infer the meaning of neutral outcomes once they determined the valence of a given cue. Upon the first encounter of a valenced outcome (reward, punishment), $Q$-values for this cue were "unmuted" and subsequently used for action selection.

We extended this base model using several bias terms. In **M2**, we added a Go bias term $b$ to the $Q$-values of making a Go response for each cue, capturing participants' overall propensity of making a Go response:

$$w(a_t, s_t) = \begin{cases} Q_t\left(a_t, s_t\right) + b & \textit{if } a = Go \\ Q_t\left(a_t, s_t\right) & \textit{else} \end{cases}$$

(3)

In **M3**, in addition to the Go bias, we added the response bias π multiplied with the cue valence $V$ (+0.5 for Win; −0.5 for Avoid; value of 0.5 arbitrarily chosen for scaling) to the $Q$-values of making a Go response:

$$w(a_t, s_t) = \begin{cases} Q_t\left(a_t, s_t\right) + b + \pi V(s) & \textit{if } a = Go \\ Q_t\left(a_t, s_t\right) & \textit{else} \end{cases}$$

(4)

Note that the cue valence was fixed and participants instructed that each cue was either a Win or an Avoid cue. Hence, no incremental learning of the cue valence was required, but the valence could be inferred as soon as the first valenced outcome was encountered. Similar to early learning from neutral outcomes, the response bias was muted (multiplied with zero) up until the first encounter of a valenced outcome and only subsequently unmuted.

In **M4**, instead of a response bias, we added a learning bias $κ$, capturing participants' tendency to use an increased learning rate for rewarded Go actions (i.e., attribute rewards to their own actions) and a decreased learning rate for punished NoGo actions (i.e., an unwillingness to attribute punishments to own inactions):

$$\varepsilon = \begin{cases} \varepsilon_0 + \kappa & \textit{if } r_t = 1 \textit{ and } a = go \\ \varepsilon_0 - \kappa & \textit{if } r_t = -1 \textit{ and } a = nogo \\ \varepsilon_0 & \textit{else} \end{cases}$$

(5)

Since learning rates were sampled in a continuous space and later inverse-logit transformed to constrain them to the range of [0, 1], the impact of $κ$ might be asymmetric after the transformation. To achieve a symmetric impact of $κ$, we first determined whether the base learning rate $\varepsilon_0$ was smaller or bigger than 0.5, then computed one half of the bias (i.e., the side closer to the range limits), took the difference between the base and the biased learning rate, and used this difference to compute the other half of the bias:

$$\varepsilon = \begin{cases} \varepsilon_0 = inv.logit(\varepsilon) \\ \varepsilon_{punished\ NoGo} = inv.logit(\varepsilon - \kappa) & \textit{if } \varepsilon_0 < 0.5 \\ \varepsilon_{rewarded\ Go} = \varepsilon_0 + \left(\varepsilon_0 - \varepsilon_{punished\ NoGo}\right) & \textit{if } \varepsilon_0 < 0.5 \end{cases}$$

(6)

$$\varepsilon = \begin{cases} \varepsilon_0 = inv.logit(\varepsilon) \\ \varepsilon_{rewarded\ Go} = inv.logit(\varepsilon + \kappa) & \textit{if } \varepsilon_0 > 0.5 \\ \varepsilon_{punished\ NoGo} = \varepsilon_0 - \left(\varepsilon_{rewarded\ Go} - \varepsilon_0\right) & \textit{if } \varepsilon_0 > 0.5 \end{cases}$$

(7)

In **M5**, we added both the response bias π and the learning bias $κ$, which provided a better fit than each bias in isolation.

In **M6**, to capture participants' overall tendency to repeat responses irrespective of the received outcome, we extended M5 by adding an intercept persistence parameter $\varphi_{INT}$ to the weight of the action performed on the last encounter of the same cue:

$$w\left(a_i, s_t\right) = \begin{cases} w\left(a_i, s_t\right) + \varphi_{INT} & \textit{if last action to same cue } s \textit{ was } a_i \\ w\left(a_i, s_t\right) & \textit{else} \end{cases}$$

(8)

Finally, in **M7**, to capture participants' higher tendency to repeat responses for Win than for Avoid cues, we added the persistence bias $\varphi_{DIFF}$, which was added to the weight of the last performed action in case of a Win cue and subtracted from this weight in case of an Avoid cue:

$$w(a_i, s_t) = \begin{cases} w(a_i, s_t) + \varphi_{INT} + \varphi_{DIFF} & \text{if Win cue and last action to same cue } s \text{ was } a_i \\ w(a_i, s_t) + \varphi_{INT} - \varphi_{DIFF} & \text{if Avoid cue and last action to same cue } s \text{ was } a_i \\ w(a_i, s_t) & \text{else} \end{cases} \tag{9}$$

We fitted models using weekly informative hyperpriors to $X_\rho \sim \mathcal{N}(2, 3)$, $X_\varepsilon \sim \mathcal{N}(0, 2)$, $X_{b,\pi,\kappa} \sim \mathcal{N}(0, 3)$ in line with previous studies [9,10]. We constrained feedback sensitivities $\rho$ to be positive using the exponential transform and learning rates $\varepsilon$ to the range [0, 1] using an inverse the inverse-logit transform. Furthermore, we constrained $\kappa$ and $\varphi_{DIFF}$ to be positive in line with the hypothesized direction of the learning and persistence biases (Fig 2B and 2C) using the $y = log(1 + exp(x))$ transform, which is $y=0$ for negative numbers, smoothly asymptotes 0 for small positive numbers, and is roughly $y=x$ for large positive numbers. Leaving these parameters unconstrained led to the same qualitative conclusions, but a slightly inferior fit. In line with recent recommendations [84], we fitted the data of each sonication condition separately.

We fitted all models using the CBM toolbox in MATLAB [85] using three steps: In the first step, we fitted the respective model to each participants' data individually using an iterative expectation-maximization algorithm that uses the Laplacian approximation for estimating the parameter posteriors. Chances of getting stuck in a local minimum were mitigated by using multiple parameter initialisations. In a second step, we used the individual fits of all participants as the starting point in a hierarchical Bayesian inference procedure. Group-level parameters were iteratively fitted using mean-field variational Bayes and served as hyperpriors for the individual participants' parameters. Thus, the parameter values of each participant were constrained/informed by the parameter values of all other participants, leading to more robust estimates and a lower chance of overfitting. We used these parameter values for all our posterior predictive checks. Finally, in a third step, we fitted all candidate models for all participants in one single step by iteratively (a) determining the probability that a given participant X was best fitted by a given model Y and (b) fitting the group-level parameters based on (only) those participants whose behavior was in fact best characterized by model Y (i.e., parameters weighted proportionally to participants' latent "model responsibility" weights). This approach allows for simultaneous hierarchical Bayesian inference (with participants constraining each other's parameter values) and random-effects model selection (allowing for the possibility that different participants are best characterized by different models). Based on this last step, we computed the model frequency and protected exceedance probability of each model and performed Bayesian model selection [37], selecting the model with the highest exceedance probability as the winning model.

We compared parameters across conditions using one-way repeated-measures ANOVAs as implemented in the ezAnova() function of the ez package in R. We computed paired $t$-tests using the t.test() function from the stats package in R and computed Hedges' $g$ as a metric of effect size by first computing Cohen's $d$ (mean of the condition difference divided by its standard deviation) and then multiplying it with the adjustment factor $J = \frac{3}{4N-5}$ ((with $N$ being the number of participants). We computed robust bootstrapped confidence intervals by, for 10,000 iterations, sampling $N$ times from the condition difference vector (with replacement), computing Cohen's $d$ per iteration, and then extracting the 2.5% and 97.5% percentiles across iterations (and multiplying them by $J$).

To validate that our best fitting model M7 could reproduce key patterns observed in the empirical data, we performed posterior predictive checks [86,87] using one-step-ahead predictions [88]. We used each participant's best fitting parameters from the hierarchical Bayesian fit and their actual choices and outcomes from the empirical data to simulate the action probabilities of each participant on each trial, which we used as synthetic data to then plot the proportions of Go response/response repetitions the same way as the empirical data.

To confirm that we could reliably measure individual differences in model parameters, we performed parameter recovery [89,90]. We first fitted a multivariate normal distribution to the best fitting parameter values of M7, sampled 1,000 new combinations of parameter values from this distribution, simulated new data based on this parameter combinations, and fitted M7 to each simulated data set. We then correlated the sampled "ground-truth" parameters to the fitted "recovered" parameters. To test whether these correlations were significantly higher than expectable by chance, we compared them against a permutation null distribution obtained by randomly permuting the assignment of ground-truth to recovered parameter values 10,000 times and computing the 95th percentile of this distribution as the upper bound of a one-sided confidence interval.

To confirm that we could reliably detect the best fitting model for a given data set, we performed model recovery [89,90]. For each of the seven candidate models, we fitted a multivariate normal distribution to the best fitting parameter values and then sampled 1,000 new parameter value combinations for each model (in total 7,000 parameter sets). We applied the following parameter constraints: $\rho < 400$ (otherwise possibility of infinitely large numbers), $\varepsilon > 0.05$ (otherwise too little learning), and $\rho$, $\pi$, $\kappa$, $\varphi_{INT}$, and $\varphi_{DIFF}$ being far enough away from zero (discarding the lowest 10% of absolute sampled parameter values). These constraints were important to ensure that each data set expressed characteristics of the respective generating model; otherwise, if one of the bias parameters was too close to zero, a more complex model would effectively reduce to a simpler model. We then simulated a new data set for each parameter combination (in total 7,000 simulated data sets) and fitted each of the seven models to each simulated data set (in total 49,000 fits). For each data set, we used the log model evidence to determine which model fitted the data best. We then computed the forward confusion matrix with the conditional probabilities (relative frequencies) of model Y emerging as a best fitting model for a data set generated by model X. We also computed the inverse confusion matrix with the conditional probabilities of model X being the generative model for a data set best fitted by model Y. To test whether the on-diagonal probabilities of these matrices were significantly higher than expectable by chance, we created a permutation null distribution by randomly permuting the log model evidences for the different fits to a given data set, computing the respective confusion matrices, and saving the on-diagonal probabilities. We again computed the 95th percentile of these distributions as upper bounds of one-sided confidence intervals.

## Supporting information

**S1 Fig. Spatial distribution of intracranial spatial peak pulse average intensity (ISPPA) during TUS for one participant.** Anatomical MRI slices are shown in sagittal, coronal, and axial views with overlaid simulated acoustic intensity maps. Red overlays indicate ISPPA values within brain tissue, scaled from 2 to 6 W/cm². Skull density is shown as a semi-transparent overlay derived from pCT Hounsfield units, ranging from 300 to 2000 HU. Orientation labels indicate left (L), right (R), anterior (A), and posterior (P).
(TIF)

**S2 Fig. Free-field acoustic simulations were conducted at an ISPPA of 30 W/cm².** As the focus is steered axially, the intensity naturally decreases as the distance from the coherent focus increases. The TPO compensates for this decrease by adjusting the power, ensuring a consistent intensity level. Therefore, along the steering range from 27.3 to 82.6 mm (measured from the transducer's exit plane), the power is regulated to maintain a constant ISPPA. The calibrated axial intensity profile plots are displayed in the figure at nine positions within the focal steering range from 27.3 to 82.6 mm. The figure is extracted from the manufacturer report. The range shown here is a calibration range used for this figure and does not represent the standard steering range of the CTX-500 system.
(TIFF)

**S3 Fig. Reaction times across sonication conditions, required action, and cue valence. A.** RTs for TUS-Sham split by required action (equivalent to accuracy given that RTs are only available for Go responses, and Go responses to Go

cues are correct, Go responses to NoGo cues are incorrect) and cue valence. Participants showed faster RTs for (correct) responses to Go cues than for (incorrect) responses to NoGo cues, $b = -0.207$, 95% CI [−0.301, 0.113], $\chi^2(1) = 18.505$, $p < .001$ and for responses to Win than to Avoid cues, $b = -0.081$, 95% CI [−0.141, −0.021], $\chi^2(1) = 6.927$, $p = .008$, reflecting a Pavlovian response bias also in RTs. This bias was slightly stronger for Go than NoGo cues, $b = -0.061$, 95% CI [−0.107, −0.014], $\chi^2(1) = 6.583$, $p = .010$. **B.** RTs per sonication condition (Sham <u>dACC</u>, aIns). There were no differences in RTs between conditions, $\chi^2(2) = 1.613$, $p = .446$. **C.** RTs split by required action and valence for all three sonication conditions. The effect of required action on RTs was significantly stronger after TUS <u>dACC</u> compared to sham, $b = -0.055$, 95% CI [−0.099, −0.010], $\chi^2(1) = 5.773$, $p = .016$, with slower errors (responses to NoGo cues) after TUS <u>dACC</u>. Furthermore, the effect of cue valence on RTs was significantly stronger after TUS-aIns compared to sham, $b = 0.037$, 95% CI [0.002, 0.071], $\chi^2(1) = 4.404$, $p = .036$, driven by faster responses to Win cues after TUS-aIns. The data underlying this figure can be found in S5 Data.
(TIF)

**S4 Fig. Pavlovian learning bias. A.** Probability of response repetitions given the previous outcome (reward vs. punishment; neutral outcomes omitted in this figure) and the previously performed action (Go vs. NoGo) for the sham condition of the empirical data. Participants show a stronger tendency to repeat responses after rewarded Go than rewarded NoGo responses, and a weaker tendency after punished Go than punished NoGo responses, reflecting a Pavlovian learning bias. **B–E.** One-step-ahead-predictions based on models M1, M3, M5, and M7. The asymmetry in outcome effects for Go and NoGo responses is only captured in M5 and M7 which feature a learning bias, but not in simpler models (M1, M3) without such a bias. The data underlying this figure can be found in S6 Data.
(TIF)

**S5 Fig. Pavlovian persistence bias. A.** Probability of response repetitions given the cue valence (Win vs. Avoid) across all trials only in the sham condition. Participants show more response repetitions for Win than Avoid cues reflecting a Pavlovian persistence bias. **B–E.** One-step-ahead prediction based on models M1, M3, M5, and M7. The difference in response repetitions between Win and Avoid cues is only sufficiently captured by a model featuring a persistence bias (M7) seen in panel E, but less so or not at all by simpler models (M1, M3, M5), in panels **B–D.** The data underlying this figure can be found in S7 Data.
(TIF)

**S6 Fig. Control models for the cue valence-specific persistence bias. A.** Simulations for Model M7 (winning model in the main text) in which cue valence directly modulates persistence. **B.** In M8, cue valence is added as a modulatory term to the prediction errors for all trials. **C.** In M9, cue valence is added to the prediction errors for trials with neutral outcomes. All models make very similar qualitative predictions regarding p(Go) per cue type, regarding p(repeat) after reward Go/ punished NoGo actions (**panels E–G**), and p(repeat) for Win/Avoid cues (**panels H–J**). Both log-model evidence (panel J) and Bayesian model selection (**panel K**) clearly favor M7 as the best model, suggesting that the behavioral finding of higher persistence for Win than Avoid cues is best captured as a modulation of the constant persistence term rather than a modulation of the prediction error term. The data underlying this figure can be found in S8 Data.
(TIF)

**S7 Fig. Parameter recovery.** When simulating 1,000 new data sets from ground-truth parameters and fitting model M7 to these data sets, the fitted parameters correlate highly with the ground-truth parameters, demonstrating the ability of the model to reliably capture individual differences in parameters. **A.** Heatmap of correlations between ground truth (x-axis) and fitted (y-axis) model parameters. Parameters were well recoverable (on-diagonal correlations: range 0.50–0.91, median 0.86) with only small off-diagonal correlations (all < |0.43|). All on-diagonal correlations were significantly higher than expectable under a permutation null distribution (1,000 permutations; 95th percentile: 0.081). **B–H**. On-diagonal

                                                                    

correlations for the feedback sensitivity ($\rho$), learning rate ($\in$), Go bias ($b$), Pavlovian response bias ($\pi$), Pavlovian learning bias ($\kappa$), persistence parameter ($\varphi_{INT}$), persistence bias ($\varphi_{DIFF}$) from simulated data.
(TIF)

**S8 Fig. Model recovery.** When simulating 1,000 new data sets from each model, fitting each data set with each model under consideration, and determining the best fitting model for each data set, the best fitting model most often corresponds to the original generative model, demonstrating the ability of reliably distinguish different models based on the data used in this experiment. **A**. Forward confusion matrix: A heatmap showing the conditional probability that data generated by a given model X (x-axis) is best fitted by model Y (y-axis). The diagonal elements show the probability of reidentifying the original generative model. All these probabilities are significantly higher than expected under a permutation null distribution (range 0.49–0.95, median 0.55; 95th percentile of permutation null distribution with 1,000 permutations: 0.171). **B**. Inverse confusion matrix: A heatmap displaying the probability that a data set best fitted by a given model X (x-axis) was in fact generated by model Y (y-axis). The diagonal elements show the probability that the best-fitting model is indeed the original generative model. All these probabilities are significantly higher than expected under a permutation null distribution (range 0.64–0.80, median 0.65; 95th percentile of permutation null distribution with 1,000 permutations: 0.171). **C**. Diagonal probabilities vs. permutation null distribution: The histogram displays the expectable on-diagonal conditional probabilities under a permutation null distribution (gray). Dashed vertical lines display the diagonal probabilities observed in the empirical confusion matrices (purple for inverse confusion matrix, green for forward confusion matrix), which are all higher than expectable under the null distribution.
(TIF)

**S9 Fig. Post hoc sensitivity analysis.** Post hoc sensitivity analysis of statistical power as a function of Hedges' g for a fixed sample size of $N = 29$ and $\alpha = 0.05$. Horizontal reference lines indicate power levels of 50%, 80%, and 95%, with the corresponding effect size values ($g \approx 0.52$, 0.73, and 0.94) indicated in the figure. Vertical dashed lines mark these effect size values.
(TIF)

**S10 Fig. Conceptual behavioral signatures of model parameters.** Conceptual illustration of the behavioral signatures associated with each model parameter. For each parameter, schematic bar plots contrast characteristic choice patterns expected under high versus low values of that parameter. Panels show how biases influence p(Go) or the probability of repeating the last response as a function of required action, previous outcome, valence, or last response. The figure is intended to provide intuition for how individual parameters map onto observable behavior rather than to depict empirical data.
(TIF)

**S11 Fig. Associations between simulated in situ pressure and subject-level sonication-induced changes in model parameters.** Scatterplots show the relationship between simulated in situ pressure (in kPa) for a given sonication condition (top row dACC, panels **A–D**; bottom row, aIns, mean of left and right aIns simulations, panels **E–H**) based on transducer and parameter coordinates recorded using Brainsight Neuronavigation during the sonications sessions, and changes in model parameters ($b$, $\kappa$, $\varphi_{INT}$, and $\varphi_{DIFF}$) between active stimulation and sham. Each point represents one participant. Red lines indicate least-squares regression fits and gray shading indicates 95% confidence intervals. Pearson correlation coefficients and two-tailed $p$-values (not corrected for multiple comparisons) are shown in the caption of each panel. Two associations reached nominal significance at the uncorrected threshold of $p < 0.05$ (**A** and **E**), but none survived correction for 8 tests (Bonferroni-corrected $\alpha = 0.00625$).
(TIF)

**S1 Table. Subject-level summary of simulated thermal and acoustic exposure metrics for each stimulation target (left and right anterior insula, dACC).** For each participant and target, Peak temperature (°C), peak pressure (kPa),

spatial-peak pulse-average intensity (ISPPA), and the transcranial mechanical index (MItc) are reported within a 10-mm-radius spherical mask centered on the default MNI coordinates used to define each target. Region labels are defined from the AICHA Atlas. These values document acoustic/thermal exposure and support safety reporting and reproducibility. (XLSX)

**S2 Table. For each participant (ID), three stimulation targets are reported: left anterior insula (l-aIns), dorsal anterior cingulate cortex (dACC), and right anterior insula (r-aIns).** Planned coordinates indicate the intended stimulation target in MNI space, while neuronavigation coordinates correspond to the final coil position recorded by the neuronavigation system, also in MNI space. Steering range (mm) refers to the scalp-to-target distance along the stimulation trajectory. Drift from sample reflects the displacement (in mm) of the coil position relative to the initially sampled position during targeting. Euclidean distance represents the three-dimensional distance (in mm) between the planned and the actual stimulated target. NA indicates unavailable data. (XLSX)

**S3 Table. Overview of results from mixed-effects logistic regression models testing for effects of TUS sonication condition on Go/NoGo choices.** Mixed-effects linear regression models were fit to participants' Go/NoGo choices with the glmer() function from lme4 package in R. P-values were determined using Wald chi-squared tests using Type-3 sums-of-squares via the Anova() function from the car package. The dependent variable was coded as Go = 1, NoGo = 0. All factors were coded with sum-to-zero coding (required action: Go = 1, NoGo = −1; valence: Win = 1, Avoid = −1; mini-block half: early = −1, late = 1) such that regression coefficients can be interpreted as standardized regression coefficients. Effects involving sonication were followed up by eliminating one factor level and refitting the model with only the two remaining levels, respectively (indented; in italics; dACC vs. Sham: dACC = +1, sham = −1; aIns vs. sham: aIns = +1, sham = −1). (XLSX)

**S1 Appendix. Adverse events and symptom reporting questionnaire.**
(PDF)

**S2 Appendix. Contraindications and safety exclusion criteria.**
(PDF)

**S1 Data. Source data.**
(XLSX)

**S2 Data. Source data.**
(XLSX)

**S3 Data. Source data.**
(XLSX)

**S4 Data. Source data.**
(XLSX)

**S5 Data. Source data.**
(XLSX)

**S6 Data. Source data.**
(XLSX)

**S7 Data. Source data.**
(XLSX)

**S8 Data. Source data.**
(XLSX)

## Acknowledgments

The authors thank the study participants for their time and commitment. The authors also thank the staff of the Brain Research Imaging Centre, University of Plymouth, for support with study procedures.

## Author contributions

**Conceptualization:** Nomiki Koutsoumpari, Johannes Algermissen, Hanneke EM den Ouden, Nadege Bault, Elsa Fouragnan.

**Data curation:** Nomiki Koutsoumpari, Siti Nurbaya Yaakub, Nadege Bault, Elsa Fouragnan.

**Formal analysis:** Nomiki Koutsoumpari, Johannes Algermissen, Siti Nurbaya Yaakub, Hanneke EM den Ouden, Elsa Fouragnan.

**Funding acquisition:** Elsa Fouragnan.

**Investigation:** Nomiki Koutsoumpari, Johannes Algermissen, Hanneke EM den Ouden, Nadege Bault, Elsa Fouragnan.

**Methodology:** Nomiki Koutsoumpari, Johannes Algermissen, Siti Nurbaya Yaakub, Hanneke EM den Ouden, Nadege Bault, Elsa Fouragnan.

**Project administration:** Nadege Bault, Elsa Fouragnan.

**Resources:** Johannes Algermissen, Nadege Bault, Elsa Fouragnan.

**Software:** Nomiki Koutsoumpari, Johannes Algermissen, Siti Nurbaya Yaakub, Elsa Fouragnan.

**Supervision:** Hanneke EM den Ouden, Nadege Bault, Elsa Fouragnan.

**Validation:** Elsa Fouragnan.

**Visualization:** Nomiki Koutsoumpari, Elsa Fouragnan.

**Writing – original draft:** Nomiki Koutsoumpari, Johannes Algermissen, Nadege Bault, Elsa Fouragnan.

**Writing – review & editing:** Nomiki Koutsoumpari, Johannes Algermissen, Siti Nurbaya Yaakub, Hanneke EM den Ouden, Nadege Bault, Elsa Fouragnan.

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
