## [Editor Report · Decision Letter 0]

11 Aug 2025

Dear Elsa,

Thank you for submitting your manuscript entitled "Ultrasound neuromodulation reveals distinct roles of the dorsal anterior cingulate cortex and anterior insula in learning" for consideration as a Research Article by PLOS Biology.

Your manuscript has now been evaluated by the PLOS Biology editorial staff and I am writing to let you know that we would like to send your submission out for external peer review.

Once your full submission is complete, your paper will undergo a series of checks in preparation for peer review. After your manuscript has passed the checks it will be sent out for review. To provide the metadata for your submission, please Login to Editorial Manager (https://www.editorialmanager.com/pbiology) within two working days, i.e. by Aug 13 2025 11:59PM.

Kind regards,

Christian

Christian Schnell, PhD

Senior Editor

PLOS Biology

cschnell@plos.org

---

## [Decision Letter · Decision Letter 1]

5 Nov 2025

Dear Elsa,

Thank you for your patience while your manuscript "Ultrasound neuromodulation reveals distinct roles of the dorsalanterior cingulate cortex and anterior insula in learning" was peer-reviewed at PLOS Biology. Please allow me to apologize for the delay in sending your decision. We were waiting to receive a report from an additional reviewer, who had promised to send a report but then stopped communicating with us. In any case, your manuscript has now been evaluated by the PLOS Biology editors, an Academic Editor with relevant expertise, and by two independent reviewers.

On light of the reviews, which you will find at the end of this email, we would like to invite you to revise the work to thoroughly address the reviewers' reports.

As you will see below, the reviewers are overall supportive of your study, but they raise a number of concerns that need to be addressed to strengthen the study further clarify some aspects of the study.

Given the extent of revision needed, we cannot make a decision about publication until we have seen the revised manuscript and your response to the reviewers' comments. Your revised manuscript is likely to be sent for further evaluation by all or a subset of the reviewers.

**IMPORTANT - SUBMITTING YOUR REVISION**

*Re-submission Checklist*

*Published Peer Review*

*PLOS Data Policy*

*Blot and Gel Data Policy*

Sincerely,

Christian

Christian Schnell, PhD

Senior Editor

PLOS Biology

cschnell@plos.org

REVIEWS:

Reviewer #1: This study uses transcranial ultrasound stimulation (TUS) in a within-subject, counter-balanced, single-blind design (N=29; sessions: sham, dACC, aIns) to probe causal contributions of dACC and anterior insula to Pavlovian learning biases during a motivational Go/NoGo task. Behavior and RL modelling indicate a double dissociation: TUS-aIns attenuates a learning bias (credit assignment asymmetries), whereas TUS-dACC increases a persistence bias (cue-valence-dependent repetition), with some non-specific increases in Go bias/persistence under both active conditions. Overall, this is an innovative and rigorous study. A few clarifications are needed especially around targeting, dose modelling, sham blinding, and non-specific effects prior to publication.

Major Comments

The central claim is a clean double dissociation yet both active conditions elevate overall Go bias and persistence relative to sham, suggesting a shared non-specific component (e.g., arousal/alerting, expectancy, auditory/somatosensory factors). To better unpack this please quantify effect sizes and confidence intervals for b and φ_INT across conditions, and state whether multiplicity-corrected results change interpretation.

Suggestions could be to re-fit a model including a session-wise "arousal/alerting" nuisance component, or use mixed models to partial out non-specific changes before testing κ and φ_DIFF contrasts.

You could report Bayes factors (or equivalently robust CIs) for the absence of κ change under dACC and of φ_DIFF change under aIns to strengthen the dissociation claim. However, the regression breakdowns (in Supplementary Data) reveal that both active conditions (dACC and aIns) produced effects across multiple cue types that are often marginally significant and overlapping. This weakens the claim of a clean double dissociation and suggests possible shared, non-specific effects. Unless anatomical targeting can be demonstrated at subregional precision (see below), functional interpretations should be tempered to emphasize relative predominance rather than strict dissociation.

Because in vivo pressure differed between subjects between conditions this needs to be accounted for. You have individualized simulations and group summaries of dose. Please exploit this by relating per-subject in situ dose (e.g., on-target P_neg, I_SPPA) to change parameters (Δκ, Δφ_DIFF, Δb, Δφ_INT vs sham). Even null correlations will bound dose-response interpretations and bolster causal specificity.

The current manuscript defines stimulation sites in broad anatomical terms ("anterior insula," "dorsal anterior cingulate cortex") but does not provide sufficient detail to establish whether specific functional subdivisions were targeted. The anterior insula comprises dorsal and ventral short gyri with distinct connectivity and functions, while the dACC can be further differentiated into dorsal, rostral, and perigenual/mid-cingulate zones (and should likely be named anterior mid-cingulate cortex). Without clarifying which subregions were stimulated, interpretation of the observed behavioral effects remains somewhat ambiguous. To strengthen anatomical specificity, the authors should include sagittal (and for dACC, also coronal) depictions of the simulated beam profiles relative to key landmarks (insula short gyri, central insular sulcus, frontal operculum; cingulate sulcus and genu).

Provide a table of per-subject MNI coordinates (x, y, z) for each target along with acoustic values (on-target pressure etc), enabling readers to evaluate whether stimulation predominantly engaged dorsal vs. ventral anterior insula, or was confined to dorsal cingulate proper vs. rostral/mid-cingulate regions. These additions are necessary to verify anatomical precision and to interpret the functional claims about double dissociations between dACC and aIns. If possible, provide group average pressure maps on an MNI template.

Acoustic metrics were extracted from 10 mm spheres centered on default MNI coordinates, which may span both dorsal and ventral anterior insula gyri. No per-subject MNI coordinates are provided, preventing evaluation of variability across participants or whether stimulation predominantly engaged dorsal vs. ventral subdivisions of aIns or different dACC subzones.

The depth of each individual target from the scalp/transducer exit plane is not reported and should be included in the individual subject tables.

The sham used bone-conduction audio to mimic TUS sounds with perceptibility validated in a separate group which is somewhat odd (why not in this group?). Side-effects appear similar across sessions but still, residual cueing cannot be fully excluded. Please clarify if participants guess condition above chance? Provide a blinding index by session. It is also odd that sensations were queried the next day? Were scalp sensations / jaw vibrations / warmth systematically queried immediately after each session (not only the next day)? Please summarize condition-wise frequencies.

Stimulation was offline then the task commenced immediately. Provide a brief rationale for the expected window of neuromodulatory after-effects at 500 kHz/10% DC (80 s) and cite prior TUS-human/NHP evidence using similar offline timings. Are effects immediate? If available, report the latency between final pulse and task onset and offset.

Minor

You show M7 (response + learning + persistence bias) best fits sham and dACC. M6 wins aIns but is nested within M7. To solidify conclusions report posterior parameter recoverability (already referenced) and cross-model generalization checks more explicitly in main text (currently in Supplementary). Add a conceptual figure mapping each parameter to behavior (much of this exists in Fig. 2; a single, simplified panel would help general readers).

The "reset/salience" framing is apt, but several paragraphs could read as over-assigning unique roles to dACC vs aIns. Since both regions co-activate and both active TUS arms shift b/φ_INT, please temper the exclusivity of the claims and emphasize "relative predominance" rather than strict functional segregation.

N=29 is reasonable for within-subject TUS but leaves limited precision for higher-order interactions. Please add a sensitivity analysis (detectable standardized effects for key parameters/contrasts) and precision plots or bootstrap CIs for Δκ and Δφ_DIFF.

The safety metrics reported are appropriate. Add the exact formula used for MItc (P_neg,tc/√f) and define whether values reported are voxelwise maxima within the ROI vs. global maxima along the beam path.

You state ISPPA ≈ 50 W/cm² in water with hydrophone validation. Please give the calibration method (hydrophone model, spatial averaging/aperture, distance), mapping from drive amplitude to on-target pressure, and the assumed attenuation law(s).

Provide neuronavigation error (FRE/TRE, mean ± SD) and any cross-session repositioning metrics.

Note counterbalancing / randomization of cue-condition assignments across sessions. Consider reporting any cue-set fixed effects.

Standardize reporting of Cohen's d or Hedges' g for all key contrasts (many are already provided in modelling section).

Use consistent "TUS-dACC / TUS-aIns" labels in text and figures.

Where χ² tests from mixed models are reported, add the test family (Type II/III) and confirm Satterthwaite vs Wald approximations. Without specifying which family and approximation was used, the reported χ² values are ambiguous and not easily reproducible

The Discussion nicely connects biases to psychopathology. Consider adding concrete predictions (e.g., altered κ or φ_DIFF in OCD/MDD) to guide future TUS trials.

Reviewer #2: The authors present a well-conducted TUS study to test the effects of sonication of the dACC and anterior insula (aIns) during learning and decision-making, measured with a well-characterised Go/NoGo learning task. The authors have interesting findings, including a dissociation between the behavioural effects of aIns TUS and dACC TUS. There is a robust imaging and sonication protocol, and good evaluation of potential side-effects, which did not differ between the sites or between active and sham stimulation (meaning blinding was sustained and the results are unlikely to be driven by expectations, a critical consideration in intervention studies). Overall, I think it is novel, informative, and cutting-edge from both a methodological and theoretical perspective.

I have a few questions/clarifications:

Abstract -

(1) The authors only introduce Pavlovian response bias in the abstract, making their initial presentation of results less clear (to me, anyway). Instead, I think the authors should briefly describe all three types of Pavlovian biases, if words allow. This would also allow a mention of the null result on response bias, which I think is important enough to merit mention in the abstract given (as I understand it) it was a central part of the authors' hypotheses.

Introduction -

(2) Similarly, an earlier discussion of the authors' original predictions of each TUS condition's effects would be helpful (they are already well-presented in the discussion, specifically the Pavlovian response bias prediction that was not borne out, but this would have been helpful at introduction stage to contextualise the results)

Methods/Results -

(3) How did the authors deal with effects of TUS condition order (e.g., sham/dACC/aINS versus someone who received dACC/aINS/sham)? This was randomised but did they authors test if there was any effect of condition order on their various outcome measures?

(4) Similarly, did the authors test or control for effects of sex and age, particularly the latter given the range of 19-59 and previous age-specific effects of certain types of brain stimulation (or was there insufficient power to test this)?

(5) Can the authors provide a few more details about their approach to multiple comparisons (only mentioned on p13 in the context of the Go bias parameter and persistence parameters)—does this mean their other parameter results all survived MC correction, or was this only applied to for post-hoc/non-primary comparisons?

(6) How did the authors plan sample size - what were the statistical power considerations?

Discussion -

(7) What are the authors' views on the directionality of the effects neurally (i.e., excitatory, inhibitory) on the aIns vs. dACC (and, more speculatively, do they think the neural effects of TUS are consistent across regions with the same protocol, or is it possible it exerts different effects on these different regions due to neuronal morphology/physiology differences)?

Signed,

C Nord

---

## [Decision Letter · Decision Letter 2]

13 Mar 2026

Dear Elsa,

Thank you for your patience while we considered your revised manuscript "Ultrasound neuromodulation reveals distinct roles of the dorsalanterior cingulate cortex and anterior insula in learning" for publication as a Research Article at PLOS Biology. This revised version of your manuscript has been evaluated by the PLOS Biology editors, the Academic Editor and the original reviewers.

Based on the reviews and on our Academic Editor's assessment of your revision, we are likely to accept this manuscript for publication, provided you satisfactorily address the remaining points raised by the reviewers. Please also make sure to address the following data and other policy-related requests:

* Please add the links to the funding agencies in the Financial Disclosure statement in the manuscript details.

* Please include information in the Methods section whether the study has been conducted according to the principles expressed in the Declaration of Helsinki.

* DATA POLICY:

Regardless of the method selected, please ensure that you provide the individual numerical values that underlie the summary data displayed in the following figure panels as they are essential for readers to assess your analysis and to reproduce it: 1F, 2EFGIJ, 3CDGH, 4BCEF, S3ABC, S4ABCDE. S5ABCDE and S6DEFGHIJ.

* CODE POLICY

Per journal policy, if you have generated any custom code during the course of this investigation, please make it available without restrictions. Please ensure that the code is sufficiently well documented and reusable, and that your Data Statement in the Editorial Manager submission system accurately describes where your code can be found. More information on our Code Policy, what and how to share can be found here: https://journals.plos.org/plosbiology/s/code-availability

* Please move the references from the supplementary information file to the main reference list, or at least make sure that the references are listed there as well. References in supplementary information files are not picked up search engines and will therefore not give appropriate credit to the authors of those studies.

* Please also move supplementary methods, results etc. into the main manuscript file. Our articles have no word count limit and we would like to make it as easy as possible for readers to find the information they are looking for.

We expect to receive your revised manuscript within two weeks.

*Published Peer Review History*

*Press*

Sincerely,

Christian

Christian Schnell, PhD

Senior Editor

cschnell@plos.org

PLOS Biology

Reviewer remarks:

Reviewer #1: The revised manuscript is substantially improved and addresses most of my prior concerns. In particular, I appreciate that the authors have appropriately softened the original "double dissociation" framing to a more cautious interpretation of functional differentiation / relative predominance, which better reflects the presence of overlapping non-specific effects across active stimulation conditions. The addition of multiplicity-corrected inference, bootstrapped confidence intervals, and nuisance-controlled analyses strengthens the modeling claims, and the revised conclusions are now more consistent with the data.

I find the core conclusion broadly supported: the computational modeling continues to suggest that anterior insula stimulation predominantly alters learning bias (κ), whereas dACC stimulation predominantly alters perseveration bias (φ_DIFF). At the same time, the supplementary cue-wise regressions still show overlapping and often marginal effects across cue types, so the current tempered language is important and should be maintained.

A few points would still improve the manuscript before final acceptance. First, the subject-level dose-response analyses were reportedly performed but omitted because they were null. Even acknowledging uncertainty in simulated in situ pressure, these analyses are directly relevant to causal interpretation and should be shown in supplementary form (e.g., scatterplots or correlation tables relating Δκ, Δφ_DIFF, Δb, and Δφ_INT to simulated pressure/intensity). Second, because the revised interpretation depends on region-specific effects surviving adjustment for shared non-specific components, the nuisance-controlled mixed-model outputs should be presented explicitly (β, CI, p-values) rather than summarized only narratively.

Overall, however, I believe the manuscript is now considerably stronger and that the main conclusions are largely supported provided the interpretation remains cautious.

Reviewer #2: I am happy with the edits the authors have made to my comment and to the other reviewer's. I congratulate them on a great contribution to the literature.

---

## [Editor Report · Decision Letter 3]

7 Apr 2026

Dear Elsa,

Thank you for the submission of your revised Research Article "Ultrasound neuromodulation reveals distinct roles of the dorsalanterior cingulate cortex and anterior insula in learning" for publication in PLOS Biology. On behalf of my colleagues and the Academic Editor, Yiheng Tu, I am pleased to say that we can in principle accept your manuscript for publication, provided you address any remaining formatting and reporting issues. These will be detailed in an email you should receive within 2-3 business days from our colleagues in the journal operations team; no action is required from you until then. Please note that we will not be able to formally accept your manuscript and schedule it for publication until you have completed any requested changes.

PRESS

We frequently collaborate with press offices. If your institution or institutions have a press office, please notify them about your upcoming paper at this point, to enable them to help maximize its impact. I have already forwarded your plans to promote your findings via a press release to my colleagues from the press department at PLOS.

Sincerely,

Christian

Christian Schnell, PhD

Senior Editor

PLOS Biology

cschnell@plos.org